# Highly metastatic claudin-low mammary cancers can originate from luminal epithelial cells

Patrick D. Rädler[1,2], Barbara L. Wehde[2], Aleata A. Triplett[2], Hridaya Shrestha[1], Jonathan H. Shepherd [3], Adam D. Pfefferle[3], Hallgeir Rui [4], Robert D. Cardiff[5], Charles M. Perou[3] & Kay-Uwe Wagner [1✉]

Claudin-low breast cancer represents an aggressive molecular subtype that is comprised of mostly triple-negative mammary tumor cells that possess stem cell-like and mesenchymal features. Little is known about the cellular origin and oncogenic drivers that promote claudin-low breast cancer. In this study, we show that persistent oncogenic RAS signaling causes highly metastatic triple-negative mammary tumors in mice. More importantly, the activation of endogenous mutant KRAS and expression of exogenous KRAS specifically in luminal epithelial cells in a continuous and differentiation stage-independent manner induces pre-neoplastic lesions that evolve into basal-like and claudin-low mammary cancers. Further investigations demonstrate that the continuous signaling of oncogenic RAS, as well as regulators of EMT, play a crucial role in the cellular plasticity and maintenance of the mesenchymal and stem cell characteristics of claudin-low mammary cancer cells.

[1] Department of Oncology, Wayne State University School of Medicine and Tumor Biology Program, Barbara Ann Karmanos Cancer Institute, Detroit, MI, USA. [2] Eppley Institute for Research in Cancer and Allied Diseases, University of Nebraska Medical Center, Omaha, NE, USA. [3] Department of Genetics, Lineberger Comprehensive Cancer Center, University of North Carolina at Chapel Hill, Chapel Hill, NC, USA. [4] Department of Pathology, Medical College of Wisconsin, Milwaukee, WI, USA. [5] Center of Comparative Medicine, University of California, Davis, CA, USA. ✉email: wagnerk@karmanos.org

Breast cancer is a heterogeneous malignancy that, based on gene expression profiling, can be stratified into five main molecular subtypes: luminal A, luminal B, HER2-enriched, basal-like, and normal-like[1,2]. An important determinant for these molecular subtypes and their clinical relevance is the expression and functionality of steroid hormone receptors and receptor tyrosine kinases, in particular estrogen and progesterone receptors (ER, PR) and the human epidermal growth factor receptor 2 (HER2, ERBB2). About 15% of all cases lack expression of ER, PR, and amplification of HER2, and most of these triple-negative breast cancers (TNBCs) are basal-like tumors[3]. A significant number of TNBCs cluster to a subordinate molecular subtype, called claudin-low, which exhibits features of mesenchymal cells. Claudin-low mammary tumors were first identified in murine breast cancer models that showed a reduced expression of genes encoding for tight junction and cell adhesion proteins (i.e., claudin 3, 4, 7, and E-cadherin)[4]. The subsequent molecular and phenotypic characterization of this subtype in human breast cancers revealed that claudin-low tumors exhibit reduced levels of differentiated luminal cell surface markers (CD24, EpCAM) and have an elevated expression of N-cadherin and vimentin. Based on the expression of CD44, CD49f, and ALDH1A1, as well as their relatively low proliferation rate, these breast cancer cells exhibit similarities to normal mammary stem cells[5–7]. The examination of 50 breast cancer cell lines showed that 7 of the 10 bona fide basal-like tumor cell lines within this collection (excluding MDA-MB-435 and HBL100) belong to the claudin-low subtype[6,8]. Among these claudin-low cell lines, MDA-MB-231 is the most commonly used TNBC human breast cancer cell model to study mechanisms of cancer invasion and metastasis.

Little is known about oncogenic drivers and the cell(s)-of-origin that give rise to claudin-low tumors. Based on their resemblance to mammary stem cells, it was proposed that this cancer subtype might originate from multipotent progenitors[9]. A common characteristic of many TNBCs, including claudin-low tumors, are mutations in Trp53 and members of the PI3K/AKT pathway[5]. This breast cancer subtype also exhibits a strong activation of RAS/MAP kinase signaling due to amplification of KRAS and BRAF, as well as loss of NF1[4,10,11]. Genomic aberrations in this pathway are further enriched in residual breast cancers following neoadjuvant chemotherapy[12], which might explain why RAS/MAPK pathway mutations are present in commonly used breast cancer cell lines that were derived from pleural effusions. Three of the seven claudin-low tumor cell lines that were identified by Prat et al.[8] have known hot-spot mutations in KRAS or HRAS. Interestingly, MDA-MB-231 cells carry mutations in BRAF[13] and NF1 (COSMIC) in addition to oncogenic KRAS, suggesting that high levels of RAS/MAP kinase signaling might play critical roles in the cellular plasticity and metastatic characteristics. This idea might be supported by recent bioinformatic studies that show that increased activation of the RAS pathway is a recurrent feature across all claudin-low breast cancers[14,15].

In this work, we performed a histological and molecular characterization of three mammary cancer models to assess whether oncogenic RAS signaling is a determinant for the genesis of triple-negative mammary tumor subtypes and whether, based on their molecular profiles, cancer cells resemble the normal epithelial subtype from which they may have originated. The collective results of this study show that oncogenic RAS signaling causes triple-negative mammary tumors that exhibit a high rate of metastasis. More importantly, we can demonstrate that luminal epithelial cells can give rise to basal-like and claudin-low mammary cancers when exogenous or endogenous mutant RAS is expressed in an epithelial cell lineage-independent manner. This study also reveals that the degree of cellular plasticity of claudin-low cancer cells is being continuously upheld by RAS-dependent and RAS-independent molecular pathways.

## Results

**Oncogenic RAS signaling initiates the development of poorly differentiated, triple-negative mammary carcinomas that have the propensity to metastasize**. We generated female transgenic mice that express the KRAS$^{G12D}$ mutant under the control of the tetracycline-responsive transactivator in the mammary gland (MMTV-tTA TetO-Kras$^{G12D}$) to determine whether oncogenic RAS signaling causes triple-negative mammary tumors (Fig. 1a). These mice also carried the TetO-H2B-GFP reporter transgene to monitor the spatially and temporally controlled expression of the MMTV-tTA in normal and neoplastic mammary epithelial cells. In adult females, the MMTV-tTA-mediated activation of the TetO-H2B-GFP responder transgene was observed exclusively in mammary epithelial cells within ducts and secretory alveoli (Suppl. Fig. 1a). Most cells that expressed nuclear GFP resided within the cytokeratin 8 (CK8)-positive, luminal epithelial subtype (Suppl. Fig. 1b, left). Expression of the MMTV-tTA-driven GFP reporter was also detected in very few isolated basal epithelial cells that were cytokeratin 14 (CK14)-positive (Suppl. Fig. 1b, right). The persistent activation of mutant KRAS under the control of the MMTV-tTA in the mammary gland was sufficient to initiate the development of palpable mammary tumors after an average latency of 160 ± 41 days (Fig. 1a, right). Four out of nine mice (45%) had overt lung metastases at the time of necropsy. Neoplastic lesions were not detected in any of the age-matched TetO-Kras$^{G12D}$ single transgenic controls. The histopathological examination of primary tumors from MMTV-tTA TetO-Kras$^{G12D}$ females showed that these cancers were poorly differentiated large-cell carcinomas with selected areas of local invasion and epithelial-mesenchymal transition (EMT) (Fig. 1b, top). All tumors were comprised of cancer cells that expressed CK8 and CK14, and a closer examination revealed that many cancer cells were dual positive for both cytokeratins (Fig. 1b, lower). This was also the case for a subset of spindle-shaped cells that showed an overall reduction in the expression of cytokeratins, in particular CK8. Mammary tumors in this model were triple-negative as they lacked expression of ERα and PR, and they did not exhibit upregulation of ERBB2 (Suppl. Fig. 2a).

The orthotopic transplantation of small tumor fragments or dissociated cancer cells into wildtype recipient mice led to a swift recurrence of mammary cancers within three weeks. All secondary tumors resembled the histopathological features of primary cancer from which they were derived (Fig. 1c, left). We also observed a selective enrichment of EMT-like cells in transplants that originated from cancer that had more extensive areas of spindle-shaped tumor cells (Fig. 1c, right). Twenty-eight of 42 recipient females (66%) had pulmonary metastases at the time of necropsy, and the co-expression of the nuclear GFP reporter is indicative for the sustained expression of oncogenic KRAS during the dissemination and metastatic growth of cancer cells (Fig. 1d). The treatment of tumor-bearing recipient mice with doxycycline (Dox) led to macroscopically complete regression of mammary tumors within 10 days (Fig. 1e), suggesting that the majority of bulk tumor cells were dependent on the sustained signaling of oncogenic RAS. Nonetheless, residual cancer cells were present in all Dox-treated recipients, even when the antibiotic was administered for more than three weeks. The histone-bound, nuclear H2B-GFP reporter remained visible in residual cancer cells within primary sites of regressed tumors and micro-metastatic lesions in the lungs of Dox-treated mice (Fig. 1f). This observation may indicate that many residual tumor cells were quiescent or slow cycling.

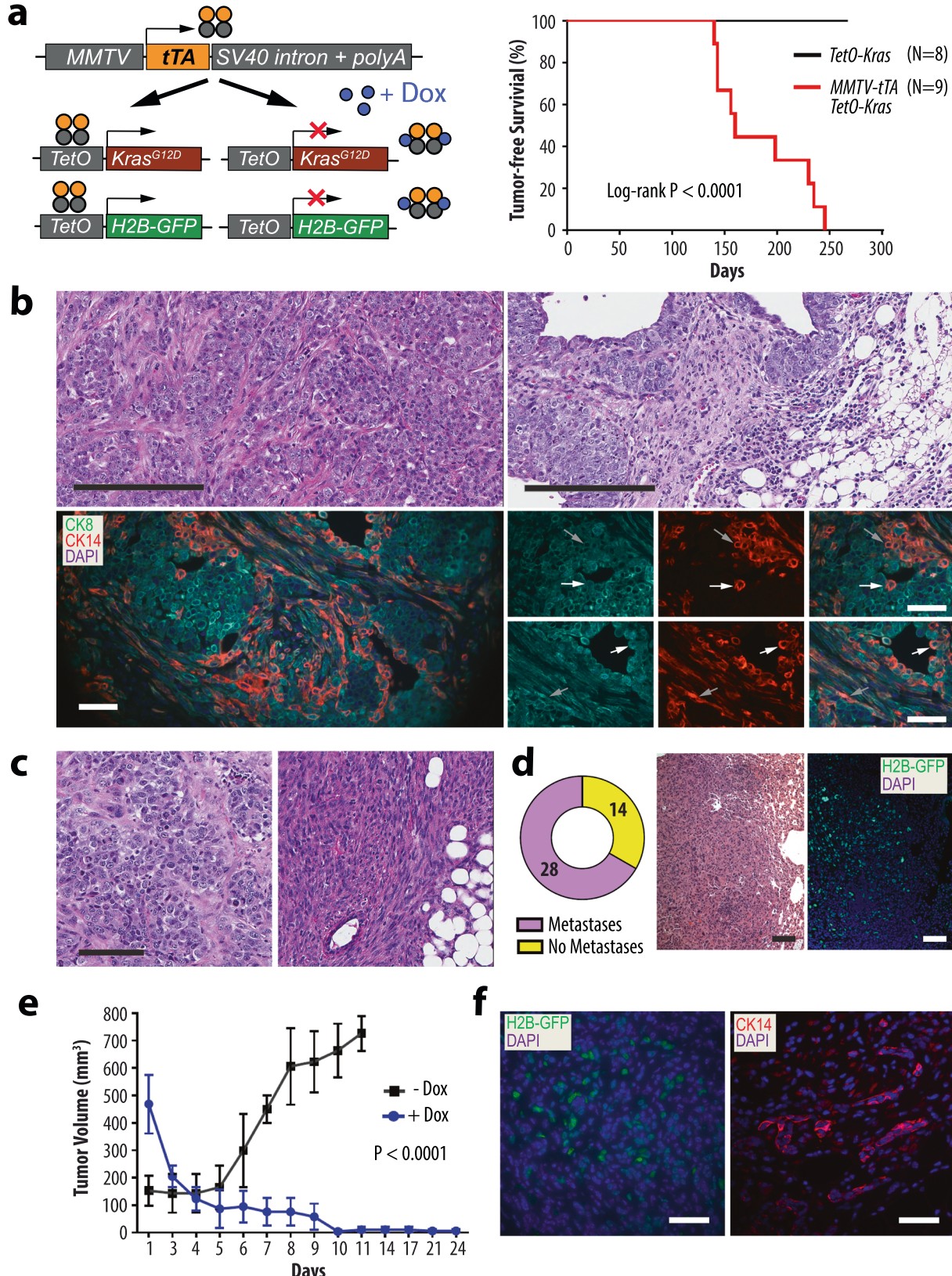

Next, we performed a microarray-based gene expression analysis on four poorly differentiated carcinomas that originated in MMTV-tTA TetO-Kras$^{G12D}$ females, including one tumor with more extensive EMT-like characteristics. The gene expression data were used for hierarchical clustering with mammary cancer profiles of a wide array of genetically engineered, as well as chemical-induced and radiation-induced mammary cancer models. The intent of this study was to determine whether mutant KRAS-expressing mammary cancers group within one or more of the nine distinct tumor classes that were defined previously[4]. The results showed that three KRAS$^{G12D}$-driven carcinomas clustered to the Class 8 tumor-type and were similar to cancers that

**Fig. 1 Expression of oncogenic KRAS under the control of the MMTV-tTA is sufficient to initiate the genesis of poorly differentiated mammary carcinomas. a** Schematic outline of the transgenes (left) and Kaplan-Meier survival plot (right) of mice that expresses oncogenic KRAS and the histone H2B-GFP reporter in a doxycycline (Dox)- controlled manner in the mammary epithelium. Statistical significance in tumor-free survival between control and experimental animals was calculated with the log-rank (Mantel-Cox) test. The resulting $P$-value was <0.0001 (Chi square = 16.86). **b** Histological sections ($N = 5$ biological replicates) and immunofluorescent staining of CK8 and CK14 ($N = 3$ biological replicates) in mammary tumors expressing oncogenic KRAS; scale bars represent 200 μm (upper) and 50 μm (lower). **c** Histological sections of transplanted tumors ($N = 7$ biological replicates); scale bar, 100 μm. **d** Rate of pulmonary metastases in tumor-bearing mice and histological section of a pulmonary metastasis with a serial section of immunofluorescently labeled H2B-GFP ($N = 3$ biological replicates); scale bar, 100 μm. **e** Tumor growth curves in wildtype recipient mice that were engrafted with mutant KRAS expressing mammary tumor cells (−Dox, $N = 3$ biological replicates). Mice with secondary tumors of about 500 mm$^3$ in volume were treated for up to 24 days with doxycycline (+Dox, $N = 4$ biological replicates) to suppress oncogenic KRAS. The data points shown represent mean values of measured tumor volumes ±SD. Statistical significance between untreated and Dox-treated mean tumor volumes was calculated with a One-way ANOVA test, resulting in a $P$-value of <0.0001 ($F = 10.92$). **f** Immunofluorescent labeling of H2B-GFP and CK14 in residual cancer cells in lung tissues of Dox-treated recipients ($N = 2$ biological replicates); bars, 50 μm.

originated in MMTV-HRAS, WAP-Myc, and WAP-Int3 transgenic females (Fig. 2, yellow arrows). Tumors within this class belong to a larger group of models that develop mainly luminal-type mammary neoplasms, including MMTV-neu and MMTV-PyMT mice. Interestingly, one MMTV-tTA TetO-KRAS$^{G12D}$-driven tumor with more extensive EMT-like, mesenchymal features clustered within the claudin-low subtype.

The collective results from the histopathological and molecular characterization suggest that expression of oncogenic KRAS under the control of the MMTV-tTA is sufficient to initiate the development of poorly differentiated mammary carcinomas that have the propensity to metastasize. While most of these neoplasms were similar to luminal-type cancers based on their gene expression profiles, they were comprised of cells that expressed both luminal and basal-type cytokeratins. In MMTV-tTA TetO-Kras$^{G12D}$ females, the progression of mammary cancers into mostly EMT-like tumors and the claudin-low molecular subtype was infrequent. Regardless of the histological and molecular characteristics, the growth and survival of the vast majority of cancer cells at primary and metastatic sites were dependent on the continuous expression of oncogenic KRAS.

**A luminal epithelial cell-specific activation of oncogenic RAS under the control of a ubiquitously active promoter leads to the development of highly metastatic, claudin-low mammary cancer.** In the mammary gland, the activation of the MMTV-tTA is limited to the epithelium and occurs predominantly in luminal cells. Since most cancer cells in the MMTV-tTA TetO-Kras$^{G12D}$ model are dependent on the continuous expression of the oncogene, it is likely that the ability of luminal-type cancer cells to transdifferentiate into a complete basal-like and mesenchymal-like state is restricted. A significantly reduced expression of the MMTV-tTA-dependent oncogenic driver in cells that undergo a mesenchymal transition would lead to their negative selection. To assess the full developmental potential of mammary cancer cells and their ability to diversify into heterogeneous lineages, we developed a mammary tumor model where the continuous expression of oncogenic KRAS is untethered from a mammary epithelial-specific promoter (Fig. 3a, left). In brief, we generated females that carry the WAP-Cre and TetO-Kras$^{G12D}$ transgenes along with a tTA knockin into the endogenous *Eef1a1* locus (EF1-LSL-tTA). A single pregnancy and lactation cycle is required to temporarily induce the expression of the WAP-Cre transgene, which is specifically activated in alveolar cells that undergo functional differentiation[16]. Within this luminal epithelial subtype, Cre recombinase excises a transcriptional *Stop* sequence (*loxP-Stop-loxP*, LSL) located between the *Eef1a1* promoter and the tTA coding sequence. This results in the constitutive activation of oncogenic KRAS, which, from this point forward, is under the exclusive control of the tTA that is driven by the ubiquitously

active *Eef1a1* locus. Following the post-lactational involution period, the EF1-tTA and TetO-driven responder transgenes remain persistently active in mostly luminal-type epithelial progenitors and their descendants[17]. A GFP-based Cre/lox reporter transgene (CAG-LSL-GFP) was used to genetically label normal and neoplastic cells that express the tTA and oncogenic KRAS.

The development of palpable mammary tumors in triple transgenic WAP-Cre EF1-LSL-tTA TetO-Kras$^{G12D}$ was contingent upon the reproductive status (Fig. 3a, right). While tumor development was not observed in age-matched, nulliparous (virgin) females, parous mice that had experienced at least one full-term pregnancy developed mammary cancer after an average latency of 226 ± 73 days. Allowing for a period of approximately 10 weeks until a female becomes primiparous, the time of tumor onset in this model was similar to MMTV-tTA TetO-Kras$^{G12D}$ mice. Thirteen of 33 (40%) mammary tumor-bearing WAP-Cre EF1-LSL-tTA TetO-Kras$^{G12D}$ females exhibited macro-metastases at the time of necropsy. In stark contrast to the MMTV-tTA-based cancer model, most WAP-Cre EF1-LSL-tTA TetO-Kras$^{G12D}$ developed EMT-type, spindle cell tumors, or mixed cancers consisting of poorly differentiated adenocarcinomas with abundant spindle-shaped cells (Fig. 3b, panels I and II). One tumor was classified as a small cell adenocarcinoma (Fig. 3b, panel III). Mammary cancers expressing mutant KRAS in a constitutive manner under the control of the EF1-tTA were triple-negative (Suppl. Fig. 2a, lanes 7–9). Following orthotopic transplantation of tumor fragments or isolated cancer cells into wildtype females, all recipients developed secondary, mammary tumors with spindle-shaped cancer cells within less than two weeks (Fig. 3b, IV). Pulmonary metastases were present in 92% of recipient animals at the time of necropsy (Fig. 3c). The CAG-LSL-GFP reporter aided in the identification of macro-metastatic and micro-metastatic tumors in the lungs, and the subsequent histopathologic examination confirmed that these GFP-positive, metastatic lesions were comprised of spindle-shaped cancer cells (Fig. 3d).

In concordance with the observed dissimilarities in their histopathological appearance, mammary tumors that express oncogenic KRAS under the control of the EF1-tTA exhibited gene expression profiles that were different from luminal-type cancers. The microarray-based gene expression analysis revealed that all mammary tumors clustered to the claudin-low intrinsic subtype and are therefore distinct from most cancers that originated in the MMTV-tTA-driven model (Fig. 2, red versus yellow arrows). As both models express similar levels of the mutant KRAS protein (Suppl. Fig. 2b), the changes in their histopathological features and gene expression profiles were not a consequence of higher transactivation of the TetO-Kras$^{G12D}$ transgene by the EF1-tTA. It was interesting to note that unlike tumors from the MMTV-tTA TetO-Kras$^{G12D}$ model, the claudin-low mammary cancers in the EF1-tTA-based model consistently showed transcriptional repression of the *Cdkn2a* locus, the resulting lack of the p19$^{Arf}$

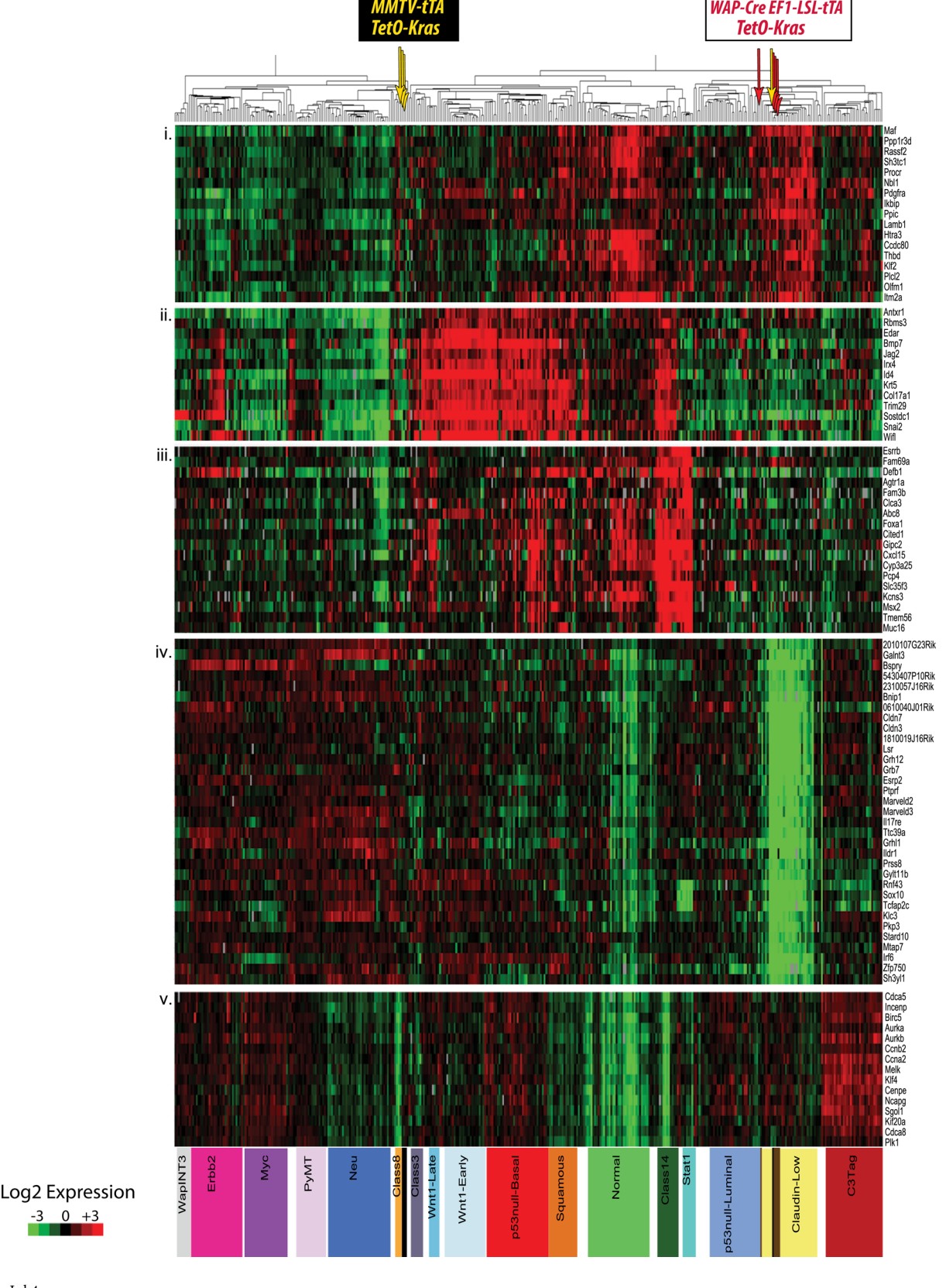

and p16$^{Ink4a}$ protein expression, and, in some cases, additional sporadic mutations in *Trp53* (Suppl. Fig. 2c, d).

**The progressive development of claudin-low tumors is preceded by a luminal-to-basal-epithelial transition.** The main distinction between the MMTV-tTA and EF1-tTA-based mammary cancer models is that the *Eef1a1* locus drives a constitutive expression of the tTA and oncogenic KRAS in neoplastic cells independent of their differentiation state, which allows them to assume diverse developmental fates. Using the Cre/lox methodology as a cell-lineage tracing tool, we previously demonstrated

**Fig. 2 The mode of expression of oncogenic KRAS determines the molecular subtype of mammary cancers.** Microarray-based gene expression and cluster analysis to compare the molecular profiles of mammary tumors that originated in MMTV-tTA TetO-Kras[G12D] (yellow arrows, $N = 4$ biological replicates) and WAP-Cre EF1-tTA TetO-Kras[G12D] (red arrows, $N = 4$ biological replicates) transgenic females with reference sets from diverse genetically engineered, as well as chemical-induced and radiation-induced mammary cancer models. Note that the TetO-Kras[G12D] transgene is under the control of the tetracycline-responsive transactivator (tTA) in both models. In contrast to the epithelial-specific expression of the tTA under the MMTV-LTR, the WAP-Cre-mediated activation of oncogenic KRAS in the luminal epithelium is under the control of the ubiquitously active EF1-tTA, which is untethered from a differentiation state and allows neoplastic cells to assume diverse developmental fates.

that a subset of functionally differentiating epithelial cells that expressed the WAP-Cre transgene during pregnancy and lactation are retained within the mammary glands of parous females[16]. In the normal gland, these parity-induced mammary epithelial cells (PI-MECs) reside within terminal ducts and function as alveolar progenitors. The vast majority of PI-MECs are luminal-type cells and distinct from CK14-positive basal epithelial cells (Suppl. Fig. 3a, left). Based on the WAP-Cre-mediated activation of the CAG-LSL-GFP reporter, it was evident that the constitutive expression of oncogenic KRAS in PI-MECs led to preneoplastic transformation and the initiation of GFP-positive mammary tumors that progressively gained invasive and mesenchymal features (Suppl. Fig. 3a right, 3b). Initially, oncogenic KRAS triggered a numeric expansion of CK8-positive PI-MECs within focal regions of the gland, but more interestingly, isolated cells with typical luminal-type appearance started to co-express CK14 (Fig. 4a, top). Groups of dual-positive cells filled the luminal space and progressively developed into basal-like preneoplastic lesions (Fig. 4a, top, Suppl. Fig. 3a, right) that progressed further into tumors that were comprised of CK14-positive cancer cells that had lost CK8 expression (Fig. 4a, middle). When primary tumor cells and their metastatic descendants subsequently had assumed mesenchymal characteristics that are reminiscent of claudin-low cancers, they exhibited a significantly reduced or complete loss of CK14 expression (Fig. 4a, lower, Suppl. Fig. 3c). Higher levels of activated ERK in GFP-labeled cancer cells, as well as preneoplastic cells in comparison to GFP-negative, untransformed epithelial cells (Suppl. Fig. 4) are indicative that the entire process of neoplastic transformation and cancer progression is controlled by mutant KRAS. Upon further examination, it was surprising to note that the mesenchymal transition process started early during neoplastic progression. We detected groups of cancer cells with typical epithelial appearance within the lumen of preneoplastic lesions that expressed higher levels of N-cadherin instead of E-cadherin (Fig. 4b, left). Along with the switch in E-/N-cadherins, these cells progressively lost expression of the tight-junction protein Occludin (OCLN, Suppl. Fig. 5). Initially, mammary tumors expressing oncogenic KRAS were comprised of pockets of cancer cells that were E-cadherin or N-cadherin positive (Fig. 4b, middle). Once these expanding tumors had completed their mesenchymal transition into the claudin-low subtype, E-cadherin-positive cells were largely absent (Fig. 4b, right). The sustained presence of GFP in tumor cells that have completed this transdifferentiation process at primary and metastatic sites is evidence that these mesenchymal-like cells originated from PI-MECs in the epithelial compartment and are therefore not tumor-associated fibroblasts (Suppl. Fig. 3b, c).

**Mutant activation of *endogenous* KRAS causes aggressive basal-type and claudin-low mammary cancers.** We generated a third cancer model to confirm that an inducible gain-of-function of endogenous KRAS in the mammary epithelium also results in the development of basal-like and claudin-low mammary cancers. This line of investigation was conducted to exclude the possibility that the initiation of these aggressive tumor types in the EF1-tTA-based transgenic model was a consequence of the exogenous

expression of the transforming oncogene. Since the MMTV-Cre-mediated activation of a conditional $Kras^{G12D}$ allele was reported to induce salivary hyperplasia[18], we developed a model where mutant KRAS is expressed from its endogenous locus in an Flp recombinase-inducible manner. In this model, the Flp recombinase is under the control of the same MMTV-LTR that was used to construct the MMTV-tTA strain. Therefore, the MMTV-Flp transgene is predominantly expressed in the luminal epithelium of the mammary gland as demonstrated in double transgenic females that carry a ROSA26-FSF-GFP reporter (Suppl. Fig. 6). In combination with the Flp-inducible $Kras^{G12D}$ knockin ($FSF$-$Kras^{G12D}$) and a single mutant $p53^{R172H}$ allele, MMTV-Flp $FSF$-$Kras^{G12D}$ $p53^{R172H}$ triple mutant females developed mammary cancer at a median latency of 97 days in an FVB/N genetic background (Fig. 5a). Mammary tumors were not observed in any of the single-transgenic littermate controls, as well as $FSF$-$Kras^{G12D}/p53^{R172H}$ females lacking the MMTV-Flp transgene while they were being maintained for approximately one year. The oncogenic activation of endogenous KRAS caused mostly EMT-like, spindle cell tumors and fewer poorly differentiated adenocarcinomas (Fig. 5b, top and middle). Despite the relatively short tumor latency, females already exhibited pulmonary metastases at the time of necropsy (Fig. 5b, bottom).

Next, we performed a PCR-based assay on 6 mammary tumors (3 per histological subtype) to validate that these cancers originated in response to the Flp-mediated activation of the endogenous mutant $Kras^{G12D}$ allele (Fig. 5c; tumor 1–3 EMT-like, 4–6 basal-like). This confirmatory experiment revealed the surprising finding that two of the three EMT-like tumors (i.e., tumor samples 2 and 3) had lost the wildtype $Kras$ allele. Consequently, the KRAS protein levels were elevated in these tumors (Fig. 5c). This line of investigation also demonstrated that the KRAS protein levels in tumors that originated in the EF1-tTA-based transgenic model (i.e., tumor samples a–c) were similar to those tumors that express mutant KRAS from the endogenous locus. This suggests that, regardless of the mode of activation, cancer cells undergo a selective process where the driver oncogene is expressed within a biologically relevant range that promotes tumor initiation and progression. Similar to the results comparing the phosphorylation of ERK1/2 between the two transgenic tumor models (Suppl. Fig. 2b), the downstream activation of these MAP kinases did not always mirror the amount of mutant KRAS protein within individual tumors (Fig. 5c).

To determine the molecular subtypes of mammary tumors that express endogenous mutant KRAS, we performed an RNA sequencing experiment on 10 primary mammary cancers (5 of each histological subtype) and conducted an unsupervised hierarchical cluster analysis together with gene reference sets from diverse mouse mammary cancer models. The results of this study established that the poorly differentiated adenocarcinomas exhibited molecular features of the basal-like subtype, and the spindle-shaped, mesenchymal-like tumors clustered closely to the claudin-low reference sets (Fig. 5d; for the complete image of the gene expression clusters please refer to Suppl. Fig. 7). The immunoblot analysis of the 6 mammary tumors that express

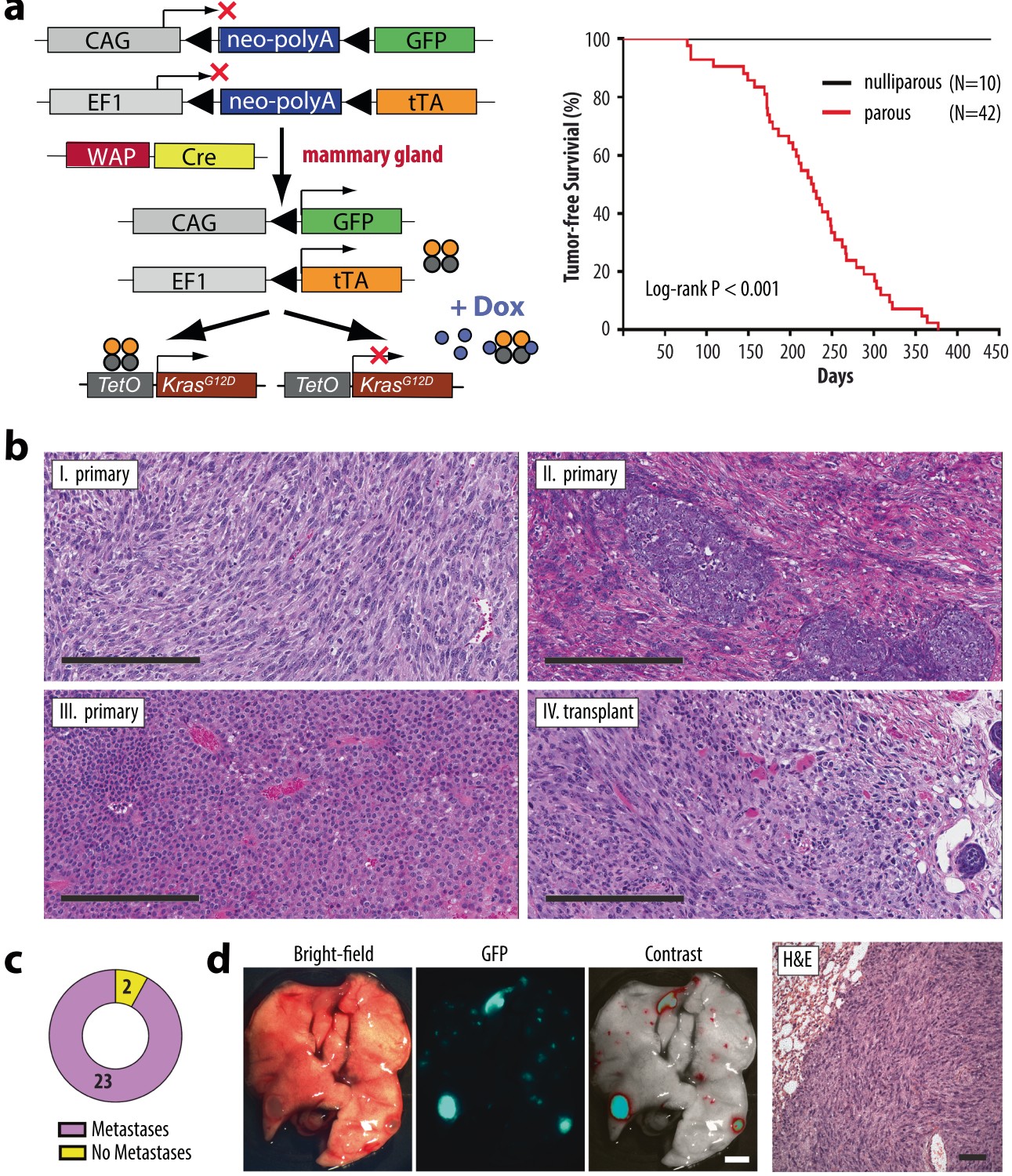

**Fig. 3 A constitutively active expression of oncogenic KRAS in luminal epithelial cells under the control of the Eef1a1 (EF1) locus-driven tTA causes the development of claudin-low mammary cancer. a** Schematic outline of the transgenes (left) and Kaplan–Meier survival plot (right) of mice that express oncogenic KRAS in a constitutively active and doxycycline (Dox)-controlled manner in luminal alveolar cells that undergo functional differentiation (i.e., WAP-Cre expression) during a full-term pregnancy. The co-activation of GFP under the control of the chicken beta-actin gene (CAG) promoter following WAP-Cre mediated recombination serves as a genetic cell lineage tracing tool during cancer progression. Statistical significance in tumor-free survival between control and experimental animals was calculated with the log-rank (Mantel-Cox) test. The resulting *P*-value was <0.0001 (Chi square = 30.57). **b** H&E-stained histological sections of primary mammary tumors (I–III, *N* = 7 biological replicates) and a transplant (IV, *N* = 5 biological replicates) from WAP-Cre EF1-tTA TetO-Kras$^{G12D}$ females; bars, 200 μm. **c** Frequency of pulmonary metastases in tumor-bearing, wildtype recipients that were orthotopically transplanted with oncogenic KRAS expressing tumors under the control of the EF1-tTA. **d** Stereoscopic brightfield and GFP images (bar, 2 mm), as well as a histological section of lungs from tumor-bearing recipient mice (bar, 50 μm) (*N* = 6 biological replicates).

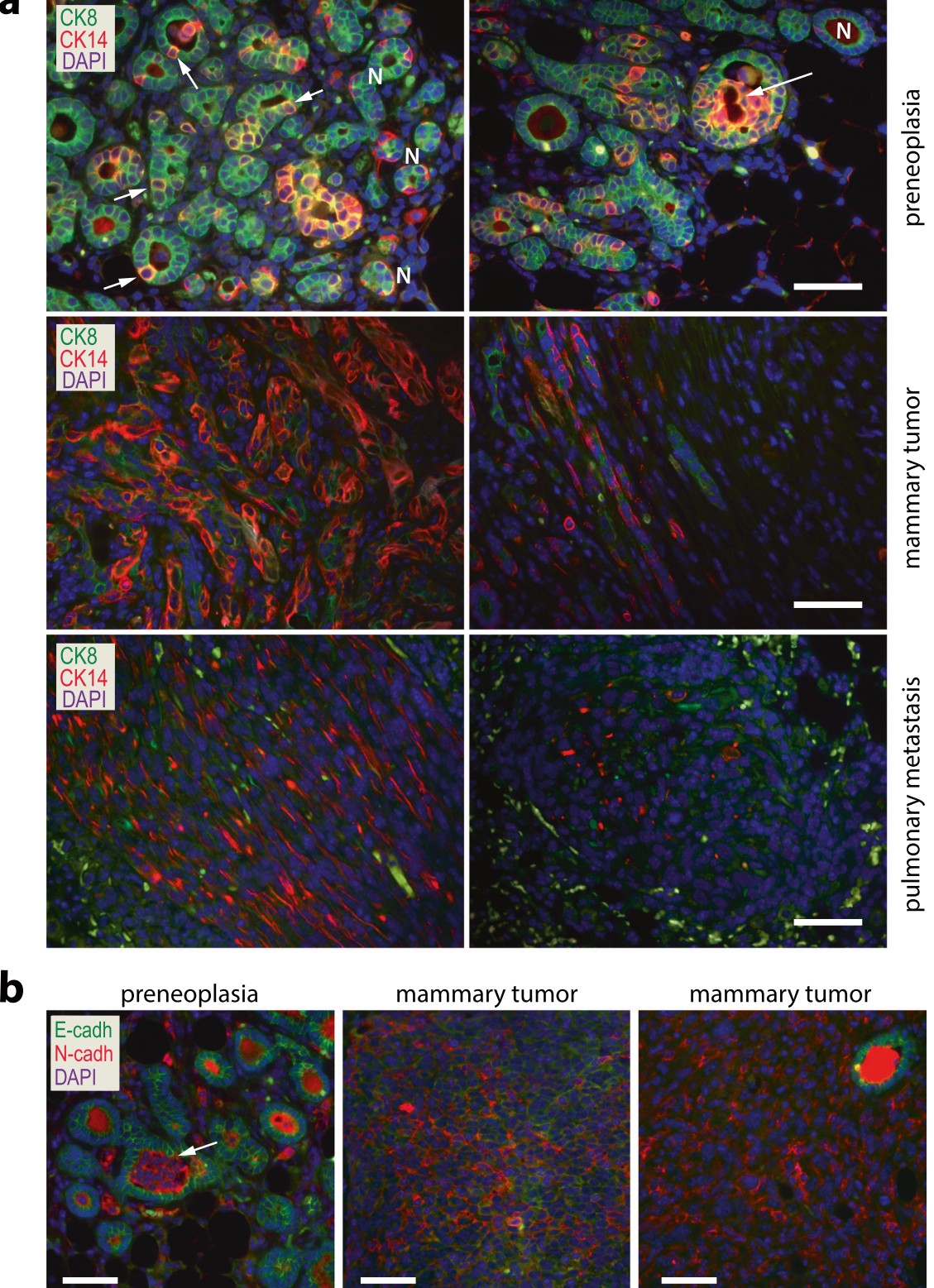

**Fig. 4 Luminal-to-basal-epithelial and mesenchymal transdifferentiation in mammary cancers that express oncogenic KRAS under the control of the constitutively active EF1-Tta. a**. Immunofluorescent staining of Cytokeratins 8 and 14 (CK8, CK14) in preneoplastic lesions ($N = 3$ biological replicates), mammary tumors ($N = 6$ biological replicates), and pulmonary metastases ($N = 6$ biological replicates) of parous WAP-Cre EF1-tTA TetO-Kras[G12D] transgenic females. **b** Immunofluorescent staining of E-Cadherin and N-Cadherin ($N = 4$ biological replicates for preneoplasia, and $N = 10$ biological replicates for mammary tumors); bars in panels **a** and **b** represent 50 μm.

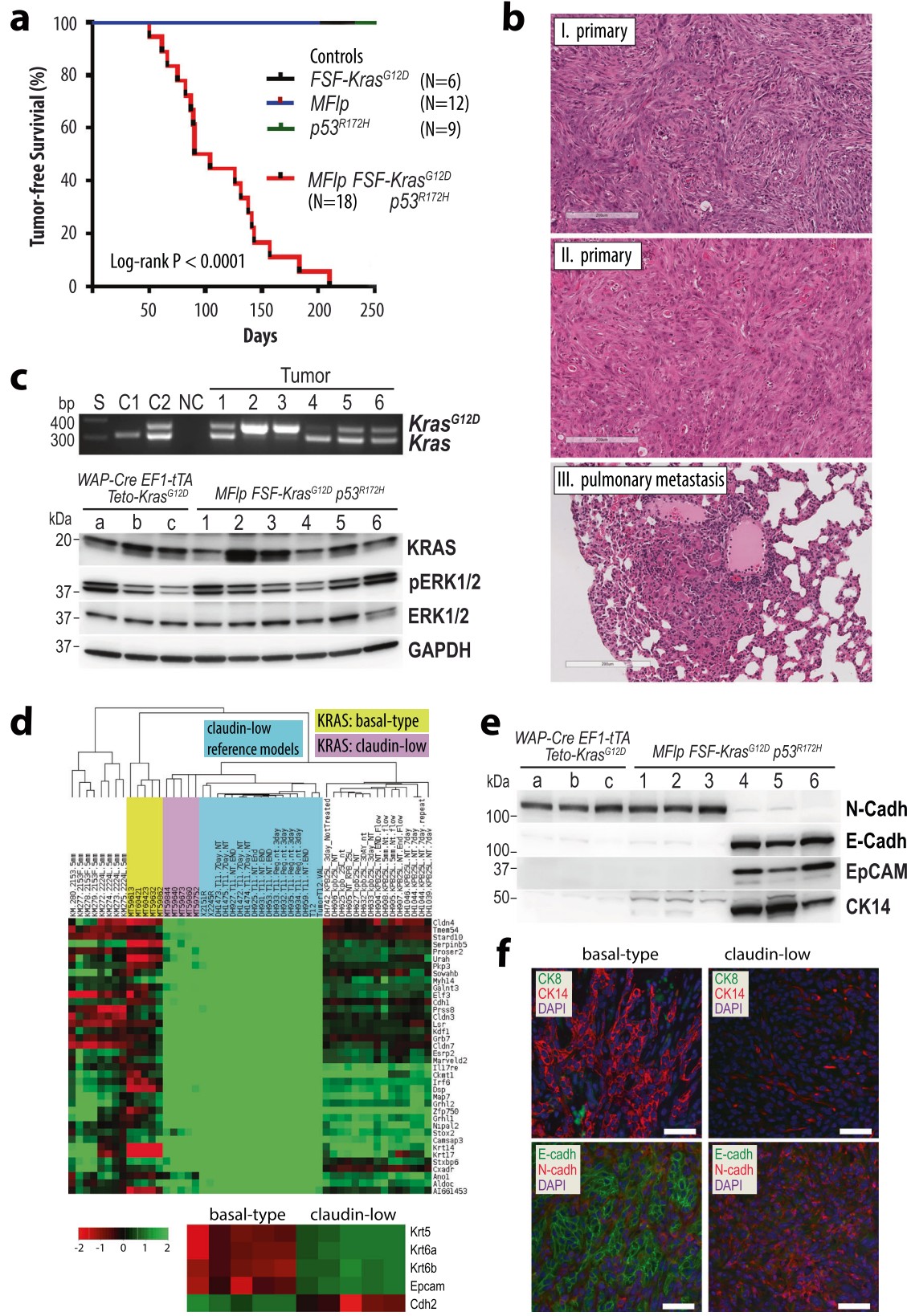

oncogenic KRAS from its endogenous locus showed that the basal-like cancers (Fig. 5e, tumors 4 to 6) express EpCAM, E-cadherin, and CK14 but lack expression of N-cadherin and CK8. All tumors lacked expression of ERα, PR, and elevated levels of ERBB2 (Suppl. Fig. 8). Like the EF1-tTA-based transgenic model (Fig. 5e, samples a to c), the claudin-low tumors from the

endogenous $Kras^{G12D}$ model (Fig. 5e, tumors 1 to 3) were N-cadherin-positive and deficient in E-cadherin and EpCAM. The differential expression of EpCAM between basal-like and claudin-low mammary tumors was validated by immunofluorescent staining (Suppl. Fig. 9). Similar to EpCAM, claudin-low tumors had lost the RNA and protein expression of CK8 and CK14, as

**Fig. 5 A gain-of-function mutation of endogenous Kras in the mammary epithelium results in the development of basal-like and claudin-low mammary cancers. a** Kaplan-Meier survival plot of mice that expresses KRAS$^{G12D}$ from the endogenous Kras gene in an Flp-recombinase-inducible manner in the presence of a mutant p53$^{R172H}$ allele (MMTV-Flp FSF-Kras$^{G12D}$ p53$^{R172H}$). Statistical significance in tumor-free survival between control and experimental animals was calculated with the log-rank (Mantel-Cox) test. The resulting P-value was <0.0001 (Chi square = 34.35). **b** Histological sections of primary mammary tumors (N = 10 biological replicates) and a pulmonary metastatic lesion (N = 6 biological replicates). **c** Top panel: PCR-based validation of the Flp-mediated excision of the Frt-Stop-Frt (FSF) sequence in the targeted FSF-Kras$^{G12D}$ allele (N = 6 biological replicates); S, DNA ladder: C1 and C2 tail DNA of mice that carry two wildtype Kras alleles (C1) or one FSF-Kras$^{G12D}$ allele (C2); NC, no DNA control. Bottom panel: Immunoblot analysis to compare the expression of KRAS and activation of MAP kinases in mammary tumors that originated in mice with exogenous mutant KRAS expression (samples **a–c**, N = 3 biological replicates) with mammary tumors expressing endogenous KRAS$^{G12D}$ (samples 1–6 corresponding to the PCR panel above, N = 6 biological replicates). Note that mice 1–6 carried one FSF-Kras$^{G12D}$ and one wildtype Kras allele. Therefore, the loss of the wildtype Kras allele in tumor samples 2 and 3 occurred somatically. **d** RNA sequencing-based gene expression and cluster analysis to compare the molecular profiles of mammary tumors expressing endogenous Kras$^{G12D}$ (N = 10 biological replicates) with reference sets from diverse mammary cancer models. The bottom panel illustrates the differences in the expression of basal-type keratins (Krt5, Krt6a/b), Epam, and N-cadherin (Cdh2) between mutant KRAS expressing tumors that exhibit similarities to basal-like or claudin-low molecular subtypes. **e** Immunoblot analysis of E-cadherin and N-cadherin, EpCam, and CK14 expression on the same mammary tumors shown in panel **c. f** Immunofluorescent staining of CK8, CK14, as well as E-cadherin and N-cadherin on sections from basal-like (N = 5 biological replicates) or claudin-low tumors (N = 5 biological replicates); bars, 50 µm.

---

well as CK5 and CK6 (Fig. 5d, bottom; 5f; Suppl. Fig. 9). Although all KRAS-driven mammary tumors had a high number of proliferating cells, we observed a significant decrease in the relative number of Ki-67-positive cells within the claudin-low subtype (Suppl. Fig. 10). Like human claudin-low breast cancers, mammary tumors of the corresponding molecular subtype in mice have more extensive immune cell infiltrates compared to basal-like mammary cancers of the same mutant KRAS-induced mammary cancer model (Suppl. Fig. 11). It is interesting to note that when mammary tumors were initiated through activation of endogenous Kras$^{G12D}$ in the presence of mutant p53, these tumors retained expression of Cdkn2a and the p19$^{Arf}$ protein (Suppl. Fig. 12). This might suggest that the loss-of-function of Cdkn2a precedes the acquisition of Trp53 mutations when Kras gain-of-function alterations are being clonally selected. Both genetic alterations are not mutually exclusive in claudin-low mammary cancer as demonstrated in the EF1-tTA-based over-expression model of oncogenic KRAS, which corresponds closely to the mutation status or lack of expression of CDKN2A and TRP53 in four of the seven human claudin-low breast cancer cell lines, including MDA-MB-231, SUM159PT, and Hs578T that carry mutant RAS.

**The cellular plasticity is controlled by persistent oncogenic RAS signaling.** The collective observations on the EF1-tTA-mediated, transgenic expression model, and mammary tumors that express oncogenic KRAS from the endogenous locus suggested that RAS signaling is a determinant for the genesis of basal-like and claudin-low molecular subtypes. Since the treatment of mice with doxycycline (Dox) led to a regression of tumors that expressed oncogenic KRAS in an EF1-tTA-dependent manner, it is evident that the proliferation and survival of most claudin-low mammary cancer cells in vivo relies on the continuous expression of the transforming oncogene (Fig. 6a). Similar to pancreatic cancer cells that express c-MYC or mutant KRAS in a ligand-controlled manner[19,20], the survival of KRAS$^{G12D}$-expressing mammary tumor cells that are being maintained in culture did not rely on the oncogenic driver (Fig. 6b). This culture model system, therefore, provided a unique opportunity to study whether the maintenance of the mesenchymal characteristics of claudin-low mammary cancer cells is dependent on the persistent expression of oncogenic KRAS. The treatment of cultured WAP-Cre EF1-tTA TetO-Kras$^{G12D}$ cancer cells with Dox for 48 to 96 h resulted in morphological changes and a significant reduction in proliferation (Fig. 6b, c). The constitutive expression of GFP from the Cre/lox reporter was used to validate these and all subsequent investigations were

carried out on pure cancer cell lines (Fig. 6b, inset). The conditional downregulation of oncogenic KRAS with Dox led to a swift dephosphorylation of downstream MAP kinases (Fig. 6d). The reduction in cleaved Caspase-3 in Dox-treated cultures indicated that cells experienced less oncogenic stress and did not initiate apoptosis in the absence of the oncogenic driver.

Next, we conducted an RNA-Seq experiment on three cell lines to determine resulting changes in their transcriptomes when the expression of oncogenic KRAS was terminated. The comprehensive analysis of their expression profiles showed that 1296 genes were downregulated and 1260 genes exhibited an increase in their mRNA expression when oncogenic RAS expression was turned off. Besides the anticipated deregulation of genes that function within the MAP kinase signaling pathway, the gene set enrichment analysis (GSEA) revealed a synchronous switch in the expression of genes that play diverse roles in biological processes such as the formation of focal adhesions and stem cell signaling (Fig. 6e), RNA biosynthesis, ECM-receptor interaction, as well as TGF-beta, HIPPO, Notch, TNF, and WNT signaling (not shown). Among the differentially expressed genes that are known to play roles in ECM signaling and focal adhesion were those that encode proteins that are being employed as markers for mammary epithelial cell lineages[21,22] (Fig. 7a). The changes in the transcriptional activation of key markers, in particular CD24, CD49f (Itga6), CD61 (Itgb3), and GATA3, were validated by immunoblot analysis and flow cytometry (Figs. 7b, 7c). Many claudin-low tumor cells that express mutant KRAS were CD24$^{low}$/CD49f$^{high}$ and lack the expression of CD61 and GATA3. The suppression of oncogenic KRAS resulted in the upregulation of CD61 and GATA3 (Fig. 7b), which are typically expressed in luminal progenitors of the normal mammary epithelium[23]. This redifferentiation process towards a more luminal epithelial fate of these tumor cells is accompanied by a substantial increase in the number of cells that exhibit high levels of CD24 and a marked reduction in CD49f (Fig. 7a, c). The partial reversal of the transdifferentiation program corresponds to our findings that luminal epithelial cells gave rise to claudin-low tumors in the oncogenic KRAS-driven, triple-negative breast cancer model.

Despite upregulation of GATA3 and significant changes in lineage marker profiles towards a luminal epithelial fate, the claudin-low mammary cancer cells did not complete a mesenchymal-epithelial transition (MET). While a subset of cells regained expression of E-cadherin in the absence of oncogenic KRAS, N-cadherin remained unchanged (Fig. 7d). With CPM values below 1, the transcription of the Occludin (Ocln) and Epcam genes was largely absent, and the downregulation of

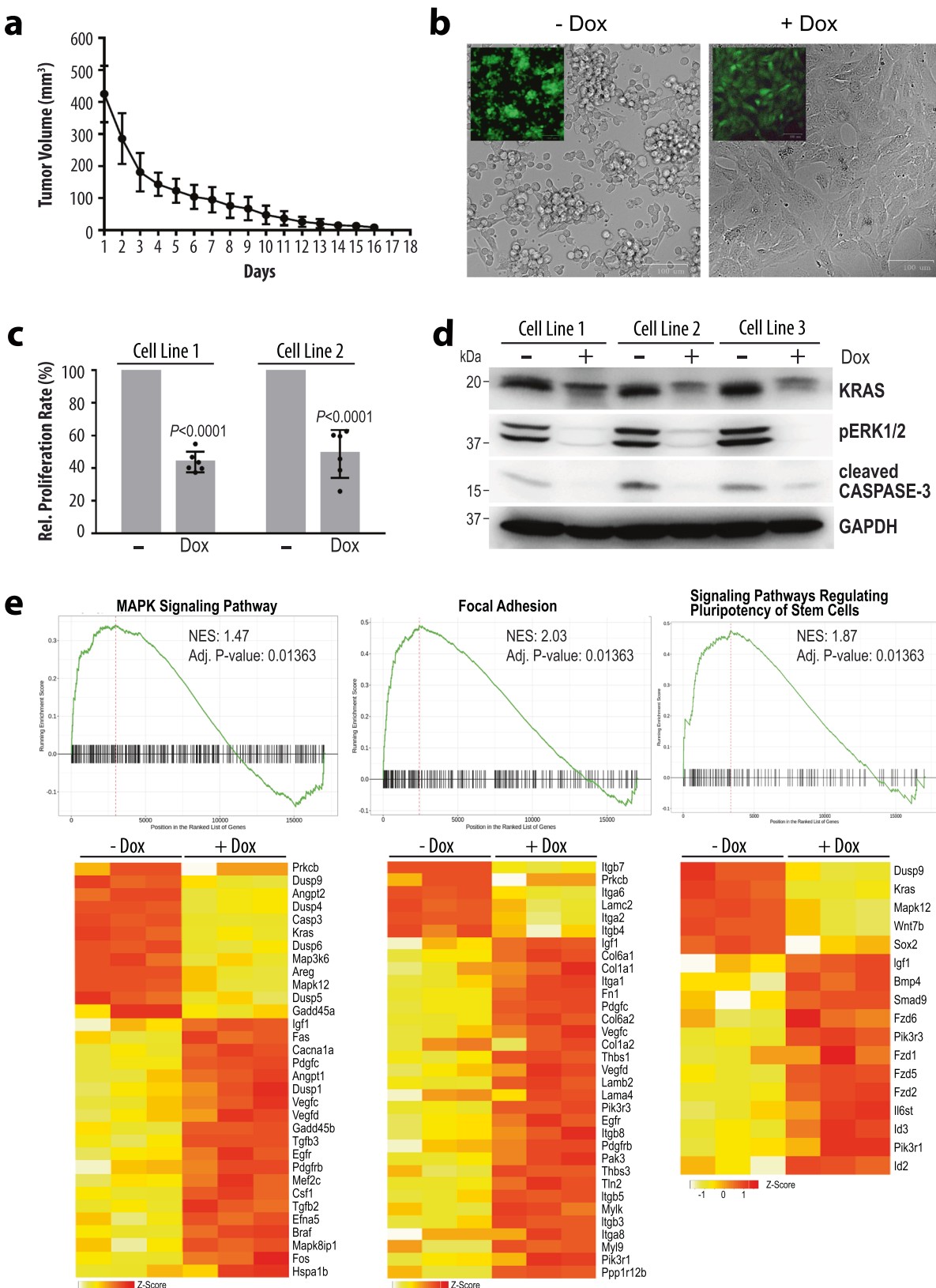

oncogenic KRAS had no effect on the mRNA expression levels of *Tjp1*, which encodes the tight junction protein ZO-1 (Suppl. Fig. 13a). Interestingly, the suppression of oncogenic KRAS led to a more than three-fold upregulation of *Snai1* and *Snai2* on the transcriptional level (Suppl. Fig. 13a), but only the SLUG protein encoded by *Snai2* was consistently higher in Dox-treated cancer

cells (Fig. 7d). The protein expression of SNAIL was unaffected by the higher transcriptional activation of its gene. Similar to *Snai1* and *Snai2*, there was also a trending increase in the RNA messages of *Zeb1* and *Twist1* (Suppl. Fig. 13a), but the steady-state levels of these proteins were not consistently deregulated and even lower in two of the three Dox-treated cell lines (Suppl.

**Fig. 6 The ligand-controlled suppression of oncogenic KRAS expression in claudin-low mammary cancer cells results in widespread transcriptomic changes. a** Regression of tumors in wildtype recipient mice that were orthotopically engrafted mammary cancer cells from WAP-Cre EF1-LSL-tTA CAG-LSL-GFP TetO-Kras[G12D] transgenics and subsequently treated with doxycycline (Dox) for 18 days after the tumors had reached about 400 mm[3] in volume. The data points shown represent mean values of measured tumor volumes ±SD. **b** Cultured mammary cancer cells from WAP-Cre EF1-LSL-tTA CAG-LSL-GFP TetO-Kras[G12D] tumors before (N = 3 biological replicates) and after (N = 3 biological replicates) treatment with Dox for 72 h. Insets show corresponding GFP fluorescent images of the same cells. **c** Decrease in proliferation rates following the silencing of oncogenic KRAS with Dox in the cell cultures shown in panel b. The mean proliferation rate of untreated cells was set to 100% and the mean relative proliferation rate of treated cells is shown ±SD, where all individual relative proliferation rates for technical replicates are represented as dots. The statistical significance of the growth rates between untreated and Dox-treated cell lines was determined with unpaired two-sided *t*-tests. The P-values were <0.0001 for both cell lines. **d** Immunoblot analysis examining the levels of KRAS, activation of MAP kinases, and activation of cleaved Caspase-3 in three independent cell cultures before (N = 3 biological replicates) and after (N = 3 biological replicates) 72 h of Dox treatment. GAPDH was used as a loading control. **e** Gene set enrichment plots of selected pathways with heat maps of corresponding genes that are selectively deregulated in their expression following the Dox-mediated suppression of oncogenic KRAS. Normalized enrichment scores (NES) and statistical significance resulted from the gseKegg function of the clusterProfiler R-package and P-values were adjusted using the default Benjamini-Hochberg method.

Fig. 13b). The collective findings support the notion that the MET program remained incomplete despite the upregulation of luminal epithelial markers, and only the SLUG protein was consistently upregulated in response to the ablation of oncogenic KRAS.

To assess the biological roles of SLUG and its closely related protein SNAIL, we performed shRNA-based knockdown experiments in the claudin-low mammary cancer cells that conditionally express mutant KRAS (Fig. 7e, Suppl. Fig. 14a). The results of this line of the investigation show that a significant reduction in SLUG led to an upregulation or sustained expression of E-cadherin in two of the three cancer cell lines when oncogenic KRAS was turned off. One of these cell lines (line #1) exhibited a co-dependency on SLUG and SNAIL as the knockdown of SLUG also resulted in the decrease of SNAIL (Fig. 7e, Suppl. Fig. 14a), and a knockdown of SNAIL in these cells was sufficient to upregulate the expression of E-cadherin in the absence of oncogenic KRAS (Suppl. Fig. 14b). The collective results suggest that while signaling of oncogenic RAS plays a significant role in maintaining molecular and cellular characteristics of claudin-low mammary cancer cells, SLUG and SNAIL may have partial roles in the maintenance of cellular plasticity independent of the driver oncogene. However, the continuous repression of E-cadherin in one of the three tumor cell lines (line 2) in the absence of oncogenic KRAS, SLUG, or SNAIL implies that, despite significant increases in ITGB3 and GATA3, the maintenance of mesenchymal characteristics of claudin-low mammary cancer cells is multifactorial and can also be controlled by other molecular mechanisms.

## Discussion

Most cancers possess characteristics that reflect their cellular origin[24], but recent bioinformatics studies showed that the classification of a tumor is not fully determined by its putative normal precursor cell[25]. Previous investigations on heterogeneous breast cancers have revealed that malignant cells can be phenotypically vastly different from the normal or premalignant cells from which they arose. A report by Lim and colleagues[26] suggested that luminal progenitors are the cellular targets for the genesis of basal-like, triple-negative tumors in women with BRCA1-associated hereditary breast cancer. The histopathological and molecular diversity among triple-negative breast cancers, which Lehmann et al.[3] stratified into six distinct subtypes, might indicate that cancer cells not only originate from different target cell populations but also assume diverse developmental trajectories.

Claudin-low breast cancers represent a subset of malignancies on the far end of the spectrum of epithelial cell plasticity where the majority of cancer cells display mesenchymal properties. It had been previously proposed that this breast cancer type originates from undifferentiated stem cells[9], but this theory had not been experimentally validated due to the lack of suitable in vivo models to study the initiation and progression of claudin-low mammary cancer. In this study, we described the generation and histopathological examination, as well as molecular characterization of three genetically engineered mammary cancer models, two of which develop primary claudin-low tumors with high incidence. We demonstrated that oncogenic RAS signaling drives the initiation and progression of triple-negative mammary cancers that originated in the luminal epithelium and evolve into highly metastatic basal-like and claudin-low mammary cancers. The collective observations suggest that metaplastic tumors of the claudin-low subtype can arise from differentiated epithelial cells that assume a developmental trajectory where they progressively gain mesenchymal and stem cell-like characteristics. This process is multifactorial and partially reversible through the targeted downregulation of the main driver oncogene.

A progressive developmental model of claudin-low mammary cancers is supported by the results of two recent bioinformatics studies on data sets from the Molecular Taxonomy of Breast Cancer International Consortium (METABRIC)[14,27]. Both reports highlighted the significant molecular diversity among claudin-low breast cancers where each tumor represents a snapshot in time of the cancer evolutionary process. As expected, basal-like tumors are significantly overrepresented in this dataset, but a portion of claudin-low breast cancers can be stratified along with other intrinsic subtypes. Similar to our in vivo cancer progression models, the bioinformatics data support the notion that epithelial cell types other than undifferentiated stem cells can give rise to claudin-low mammary cancers. The *continuous developmental model* proposed by Fougner et al.[27], which suggests that a pure claudin-low subtype may gradually emerge, is supported by the longitudinal analyses in our genetically engineered cancer models. The degree of cellular plasticity is dependent on the activation of an EMT process[14], which, as we demonstrated in this study, is reversible. The mesenchymal and stem cell features of claudin-low cancer cells in the oncogenic RAS-induced mammary cancer model are being continuously upheld by the oncogenic driver, as well as RAS-independent molecular pathways that include classical transcription factors that control EMT. It should also be noted that the Cre/lox-based cell lineage tracing of oncogene-expressing cells within preneoplastic lesions, primary and metastatic tumors, as well as secondary cancers in transplanted recipients, confirms that the spindle-shaped, metaplastic cancer cells are true cellular descendants of transformed epithelial cells and not cancer-associated fibroblasts.

Although tumor subtype-specific genomic alterations have not been identified in claudin-low breast cancers, a high activation of

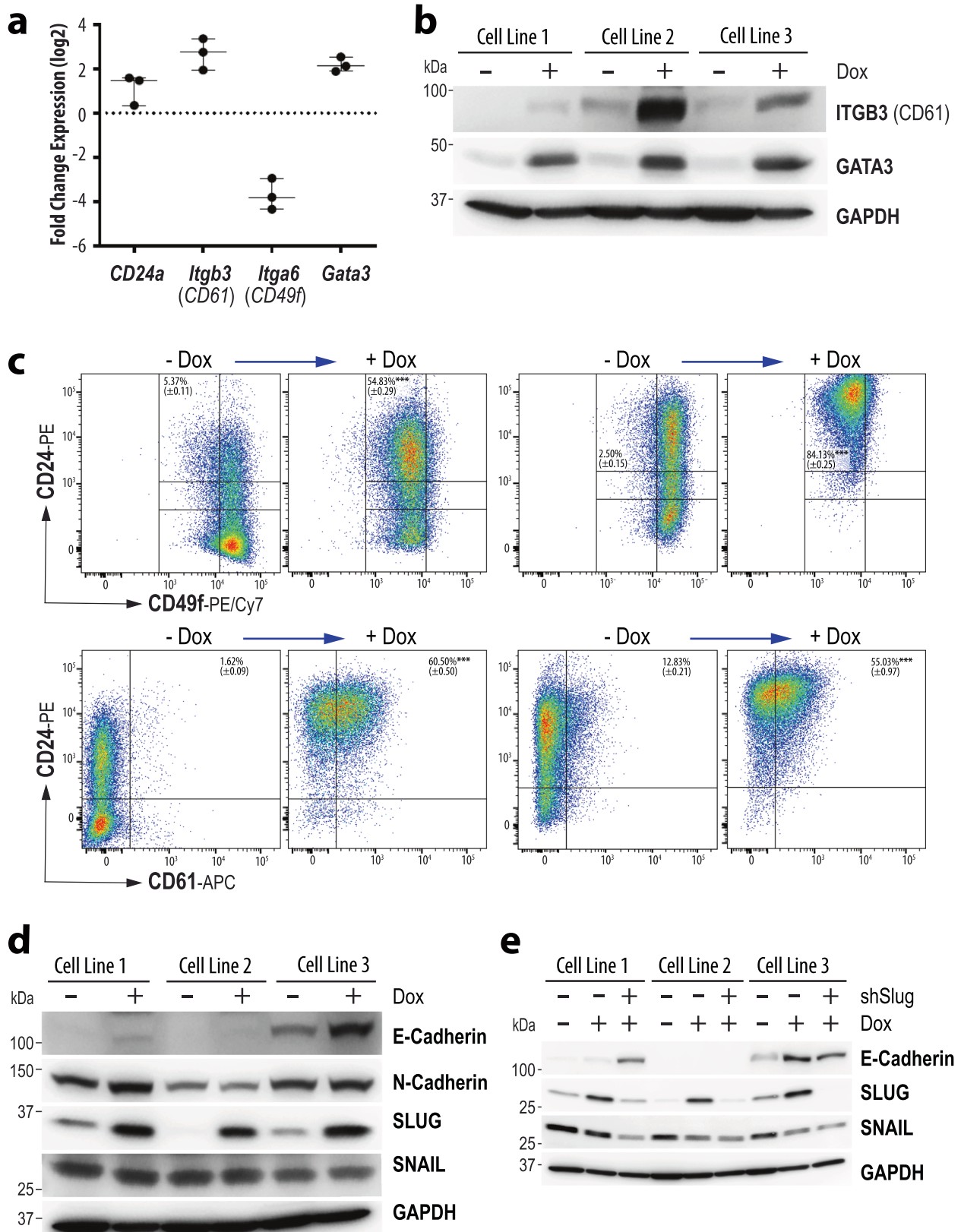

the RAS/MAPK pathway and mutations in *Trp53* seem to play critical roles. All seven *bona fide* claudin-low tumor cell lines reported by Prat et al.[8] carry mutations in *Trp53*, and half of these lines have activating mutations in one of the RAS genes in addition to other genomic alterations within the MAPK signaling pathway and their effectors, including PI3K. In their recent publication, Pommier et al.[14] singled out the activation of the RAS/MAPK pathway as one of the most recurrent features across all claudin-low breast cancers, and they reported that RAS signaling is exceptionally high in a subset of these cancers that possess stem cell features. Our work provides evidence that oncogenic RAS signaling is a direct determinant for the initiation

**Fig. 7 Oncogenic RAS signaling controls the cellular plasticity of claudin-low mammary cancer cells. a** Graphic illustration of the changes in mRNA expression of genes defining mammary epithelial cell lineages after the suppression of oncogenic KRAS in claudin-low mammary tumor cells. **b** Immunoblot analysis to validate the upregulation of the luminal epithelial lineage and progenitor markers CD61 and GATA3 following Dox-mediated suppression of oncogenic KRAS for 72 h in three independent cell lines ($N = 3$ biological replicates). GAPDH was used as a loading control. **c** Flow cytometric analysis of CD24, CD49f, and CD61 in claudin-low mammary tumor cells before and after Dox-mediated suppression of oncogenic KRAS for 72 h. **d** Immunoblot analysis of E-cadherin and N-cadherin, SLUG, and SNAIL ($N = 3$ biological replicates for each treatment group). **e** Immunoblot analysis to assess changes in E-cadherin expression following shRNA-mediated knockdown of SLUG and Dox-mediated suppression of oncogenic KRAS ($N = 3$ biological replicates).

---

of triple-negative mammary tumors that progress towards the claudin-low subtype. Although this entire developmental process is multifactorial and requires additional mutations, epigenetic alterations associated with changes in the expression in tumor suppressors (e.g., *Trp53, Cdkn2a*) and EMT-related transcription factors, primary and metastatic tumor cells with mesenchymal characteristics were dependent on the persistent expression of oncogenic RAS for tumor cell growth and survival in vivo. Together with reports that human claudin-low cancer cell lines are significantly more sensitive to MEK inhibitors[14], our findings may have important implications for the development of regimens to treat this triple-negative breast cancer subtype. To increase the effectiveness of such strategies, it will be important to elucidate the mechanisms by which a subset of cancer cells survive and remain dormant at primary and metastatic sites as shown in our genetic models. To accurately detect residual disease or circulating tumor cells as part of these therapeutic regimens, it will be critical to developing appropriate biomarker assays that take into consideration that the mesenchymal features and expression of extracellular stem cell markers may change significantly since these traits are dependent on the oncogenic driver.

## Methods

**Genetically modified mouse strains**. Our team constructed the EF1-LSL-tTA knockin [*Eef1a1*tm1(tTA)Kuw], as well as the MMTV-tTA [Tg(MMTV-tTA)25754Kuw] and WAP-Cre [Tg(Wap-cre)11738Mam] transgenic lines[17,28,29]. A detailed description of the generation and analysis of the MMTV-Flp line will be provided elsewhere. The CAG-LSL-GFP reporter strain was kindly provided by Dr. Miyazaki (Osaka University)[30]. The TetO-Kras[G12D] [Tg(tetO-Kras2)12Hev/J][31], TetO-H2B-GFP (Tg(tetO-HIST1H2BJ/GFP)47Efu/J)[32], Rosa26[CAG-FSF-GFP] [Gt (ROSA)26Sortm1.2(CAG-EGFP)Fsh][33], and FSF-Kras[G12D] [B6(Cg)-Krastm5Tyj/J][34] mouse strains were obtained from The Jackson Laboratory. Mice with the LSL-*Trp53*R172H/+ knockin allele[35] were provided by the NCI Mouse Repository. The LSL-*Trp53*R172H/+ allele was passed through the germline of MMTV-Cre, allele A [Tg(MMTV-cre)1Mam] transgenic females[29,36] to generate the *Trp53*R172H mutant allele described in this study. The transgenes were carried in the FVB/N genetic background. The FSF-Kras[G12D], *Trp53*R172H/+, and Rosa26[CAG-FSF-GFP] alleles were backcrossed seven times with FVB/N wildtype mice. The PCR primers to genotype transgenes and genetically engineered alleles are provided in Supplementary Table 1, and additional specific animal procedures are provided in Supplementary Materials and Methods. Further information and requests for the EF1-LSL-tTA knockin mouse model and the MMTV-Flp transgenic strain should be directed to the Lead Contact, Kay-Uwe Wagner. All mice were housed under pathogen-free conditions in micro-isolator cages on a 12/12-h light/dark cycle. This work was conducted in accordance with the recommendations in the Guide for the Care and Use of Laboratory Animals of the National Institutes of Health. We have complied with all relevant ethical regulations. The animal study protocols were approved by the Institutional Animal Care and Use Committee of the Nebraska Medical Center and Wayne State University.

**Histologic analysis and immunostaining**. Fresh tissue samples from whole mounts of inguinal mammary glands and lungs were prepared by spreading them on glass slides. These unfixed specimens were examined with a Discovery.V8 fluorescence stereoscope (Carl Zeiss, Inc.) for GFP expression. For histological examination, tissues were fixed overnight at room temperature in 10% buffered formalin (Fisher Scientific Company) and stored in 70% ethanol prior to paraffin embedding and sectioning. Slides were stained with Hematoxylin and Eosin (H&E) for routine histology and scanned on a brightfield slide scanner from Leica Microsystems, Inc. A recent protocol for immunohistochemistry or immuno-fluorescent staining on paraffin-embedded specimens was described by Wehde

et al.[37]. The list of primary and secondary antibodies for immunostaining is provided in Supplementary Table 2 *in the Supplementary Materials and Methods* section. Stained slides were examined with an Axio Imager microscope (Carl Zeiss, Inc.) equipped with a SPOT FLEX camera (Diagnostic Instruments, Inc.). For quantification, Ki67-positive nuclei or CD3-positive cells were counted in three representative images (magnification of ×400) of non-overlapping tumor regions. The relative number of Ki67-positive or CD3-positive cells in each of these images was determined with the software Fiji (v2.0.0; https://fiji.sc/) by counting the Ki67-positive nuclei or CD3-positive cells and the total number of DAPI-stained nuclei. The accuracy of the software was confirmed by manual counting of select images. A paired Student's *t*-test using Prism6 (v6.07; GraphPad Software, La Jolla, CA) was performed to assess statistically significant differences in cellular proliferation. The resulting data was visualized as box and whisker plots with an indication of the minimum and maximum value, as well as the first quartile, median, and third quartile. When n ≤ 10 all data points are indicated.

**Cell culture, doxycycline treatment, knockdown of SLUG and SNAIL**. Mammary cancer cells were derived from enzymatically dissociated mammary tumors and cultured in Dulbecco-modified Eagle medium (DMEM/F12) supplemented with insulin, epidermal growth factor, tetracycline-free fetal bovine serum, penicillin/streptomycin, and gentamicin. For doxycycline (Dox) treatment, the cells were split into 10 cm cell culture dishes, and 20 µg Dox per 10 ml of media was added to the cells at 0 h. Cells were washed with 1× PBS and Dox-containing media was replaced after 48 h. After 72 or 96 h, the cells were washed and collected for immunoblotting and flow cytometry using 0.05% Trypsin-EDTA or by scraping cells in ice-cold 1× PBS. The construction or source of the lentiviral gene transfer vectors to express shRNAs against SLUG and SNAIL are described in the *Supplementary Materials and Methods*. The proliferation rates of cell lines were determined by plating $1 \times 10^4$ cells into 15 wells of 24-well cell culture plates (Corning #3524). After 24 h, 48 h, 72 h, 96 h, and 120 h, cells were collected from three wells per timepoint and the total number of cells for each well was determined by manually counting cells with a hemocytometer. To illustrate changes in relative proliferation rates between Dox-treated cell lines and their untreated isogenic controls, the proliferation rates of the untreated controls were normalized.

**Western blot analysis**. Detailed experimental procedures for immunoblot analysis were described elsewhere[37]. The composition of blotting reagents and buffers is described in the *Supplementary Materials and Methods*. A list of primary and secondary antibodies for immunoblotting is provided in Supplementary Table 3. Chemiluminescence and brightfield images of the blots with size markers were taken using a D1001 KwikQuant Imager (Kindle Biosciences, LLC).

**Flow cytometry**. The flow cytometric analysis for the expression of mammary cell markers (CD61, CD24, CD49f) was carried out by incubating the cells with primary antibodies against the respective antigen in FACS buffer (1× PBS supplemented with 2% bovine serum albumin) for 25 min at 4 °C. Subsequently, cells were washed once in ice-cold FACS buffer to remove excess antibodies. PE, PE/Cy7, and APC-conjugated monoclonal antibodies against CD24 (BD Pharmingen, 553262; 1:70 dilution), CD49f (BioLegend, 313621; 1:40 dilution), and CD61 (Thermo Fisher Scientific, MCD6105; 1:100 dilution) were used. The flow cytometric data was acquired on a BD Biosciences LSRII and analyzed using the (FACSdiva 8.0.1 software). Compensation was performed at the time of acquisition using BD FACSdiva software. The gating strategy for the flow cytometric analysis is shown in Suppl. Fig. 15.

**Microarray and RNA-sequencing**. Total RNA was extracted from fresh or flash-frozen mammary tumors or cultured cancer cells using the RNeasy Mini Kit (QIAGEN). The concentration of the RNA was determined on a NanoDrop spectrophotometer, and the integrity of the RNA was validated using gel electrophoresis. RNA was processed and labeled for microarray analysis as described[38] and hybridized to Agilent Mouse 430 A 2.0 arrays. Sample standardized and gene median-centered data from samples were combined with 385 published microarray analyses from mouse mammary tumors from genetically engineered mouse models (GEMMs). Samples were hierarchically clustered and classified using the centroid method in Cluster3.0 and the set of ~1800 intrinsic murine genes described by Pfefferle et al.[38].

The RNA expression library construction and next-generation sequencing were performed by Novogene. The quality of sequenced reads was determined using FastQC (v0.11.9; http://www.bioinformatics.babraham.ac.uk/projects/fastqc). For differential expression analyses, the 150 base pair paired-end reads were mapped to the mm10 mouse reference genome with Rsubread (v2.0.0)[39]. Transcript abundance was determined using featureCounts from the Rsubread package. Low abundance transcripts (cpm < =5) that occurred in more than half of the samples were excluded from subsequent analyses. The R package edgeR (v3.28.0)[40] was used to normalize the transcript counts and to perform differential expression analysis of paired samples (i.e., expression analyses of the same cells on and off doxycycline). Transcript abundance was estimated in counts per million (cpm). Genes that showed differential expression by more than 2-fold and an FDR (False Discovery Rate) of below 0.001 between the sample groups were considered as significantly deregulated. The gene set enrichment analysis was performed with R-function gseKEGG and plotted with the R-function gseaplot (package: clusterProfiler v3.0.4)[41]. The corresponding heatmaps were created by log2 transforming the count data and by extracting the significantly deregulated genes that belong to the corresponding pathways. Heatmaps were plotted with the R-function heatmap.2 (package: gplots v3.0.3) (http://cran.r-project.org/web/packages/gplots/index.html). The Broad Institute's Integrative Genomics Viewer (IGV; v2.7.2) was used to visualize the expression of individual genes and their exons.

To perform a hierarchical cluster analysis of mammary tumors from MMTV-Flp *FSF-Kras*[G12D] females together with gene reference sets from diverse mouse mammary cancer models, RNA-sequencing (RNAseq) data was upper quartile normalized and combined with previously published RNAseq data from a variety of murine mammary tumor and normal samples (GSE118164 and GSE124821; see Supplementary Data 1). Samples were clustered by the murine intrinsic gene list using the centroid method in Cluster3.0 (v3.0)[42].

**Statistical analysis**. All graphic illustrations and statistics were performed with GraphPad Prism 6 software (GraphPad Software, Inc., La Jolla, CA). All reported measurements were taken from distinct samples. Unless otherwise indicated in the figure legends, experimental data are shown as mean values ± SEM or mean ± SD. For statistical analysis, the data were assessed for normality followed by an unpaired Student's *t*-test, Mann-Whitney-Wilcoxon test, or one-way ANOVA and Tukey's multiple comparison test. The tumor-free survival distributions between animals were assessed using the logrank test. $P$ values <0.05 (∗), <0.01 (∗∗), or <0.001 (∗∗∗) were considered statistically significant.

**Reporting summary**. Further information on research design is available in the Nature Research Reporting Summary linked to this article.

## Data availability

The RNA sequencing data were deposited in the Gene Expression Omnibus (GEO) under accession number GSE157333. Source data are provided with this paper. The Supplementary Data 1 table provides a detailed list of RNA Sequencing data sets from reference mammary tumor models. The GEO data sets associated with the Supplementary Data 1 are GSE118164, GSE124821, and GSE148482. Microarray data sets from reference mammary tumor models are available under GEO accession numbers GSE3165, GSE8516, GSE9343, GSE14457, GSE15263, GSE17916, GSE27101, and GSE42640. The raw data from these sources are associated with Figs. 2, 5D, 6E, and Supplementary Fig. 7. All other data supporting the findings of this study are available within the article and its supplementary information files and from the corresponding author upon reasonable request. A reporting summary for this article is available as a Supplementary Information file. Source data are provided with this paper.

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

## Acknowledgements

The authors thank Kerry Vistisen and Yong Li for animal maintenance and genotyping. We are grateful to Melissa Anders-DiPonio and members of the Animal Model and Therapeutics Evaluation Core Facility at the Karmanos Cancer Institute (KCI) for the preparation of histologic sections. Histologic services were also provided by the KCI Biobanking and Correlative Sciences Core Facility. Dr. Jessica Back, co-director of the KCI Microscopic Imaging and Cytometry Resources Core Facility, assisted in the acquisition and presentation of the flow cytometry data. The KCI core facilities are supported by the Public Health Service grant CA022453. The mammary cancer transcriptome analysis was funded, in part, by the METAvivor Research Foundation. Financial support by Public Health Service grants CA117930 and CA202917 (to K.-U. W.) was imperative to maintain selected mutant mouse strains that were used in this study. B.L.W. and P.D.R. received graduate fellowships through the Cancer Research Training Program of the University of Nebraska Medical Center (CA009476). C.M.P. was supported by the Public Health Service grants CA14876 (R01) and CA58223 (P50, NCI Breast SPORE), as well as a grant from the Breast Cancer Research Foundation. J.S. was partially supported by NCI F32CA228326. The funders had no role in study design, data collection, and analysis, decision to publish, or preparation of the manuscript.

## Author contributions

K.-U.W. formulated the overarching research goals and aims, supervised the research, and wrote the manuscript. P.D.R. drafted the institutional animal study protocol and performed most in vivo and cell culture experiments, immunoblots, immunofluorescent staining techniques, as well as computational analyses of RNA sequencing data sets. B.L.W. executed the mammary tumor cell transplantation experiments and tumor regression studies. H.S. assisted with genotyping, cell culture, and tissue processing. A.P, J.S., and C.M.P. conducted the microarray studies, cluster analyses of RNA-sequencing data sets, and provided instructional assistance to P.D.R. for computational processes. R.D.C. performed the H&E staining, whole-slide scanning, and comparative analysis of the histopathology of mammary tumors that originated in the oncogenic KRAS overexpression models. H.R. assisted with digital imaging of tumor tissue sections from mice expressing endogenous and exogenous mutant KRAS. A.A.T. performed molecular analyses, helped to develop genotyping protocols, and edited the manuscript.

## Competing interests

C.M.P. is an equity stockholder and consultant of BioClassifier LLC; C.M.P. has also listed an inventor on patent applications for the Breast PAM50 Subtyping assay. The remaining authors declare no competing interests.
