## [Peer Review File · Nature Communications]

REVIEWER COMMENTS

Reviewer #1 (Remarks to the Author):

In this manuscript the authors have generated three different mouse models of mutant KRAS (G12D) that gave rise to poorly differentiated mammary tumors and demonstrate oncogenic RAS signaling drives triple-negative basal-like and claudin-low mammary cancers. Claudin-low tumors are characterized by mesenchymal features, high levels of immune cell infiltration, low expression of cellular junctions and lower genomic instability, mutational burden and proliferation levels. The claudin-low phenotype was initially identified in cell line models and several cell lines have activating mutations in KRAS or HRAS. In the first model, mutant Kras is under the control of tetracycline transactivator in the ubiquitously active mammary gland MMTV promoter that gives rise to luminal and claudin-low subtype tumors by microarray. They demonstrate that the tumors are dependent of KRAS, as doxycycline leads to complete tumor regression. In the second model the authors constitutively express oncogenic KRAS in luminal alveolar cells that undergo functional epithelial differentiation during a full-term pregnancy using the mammary epithelial-specific promoter (WAP-Cre expression) and give rise to spindle cell claudin-low tumors that more frequently metastasize to the lungs. In the third model, they demonstrate inducible mutant KRAS expressed from its endogenous locus in a Flp recombinase-inducible manner in mammary epithelium with mutant p53 also results in the development of basal-like and claudin-low mammary cancers. They demonstrate the basal-like tumor express KRT14, EpCaM and E-Cadh, while claudin-low tumors express N-Cadh. Using cell line models derived from WAP-Cre Kras tumors, the authors identify genes that change in expression when mutant Kras is turned off are enriched in MAPK, focal adhesion and stem cell pathways. They validate changes in CD49f, CD24 and CD61 by flow cytometry and SLUG and E-Cadh by immunoblot. The shift to a more luminal epithelial fate when mutant KRAS is removed demonstrates oncogenic Ras as regulator of EMT and in the cellular plasticity. Overall, this manuscript provides several novel murine models and insight into the cellular origin of claudin-low breast cancer.

Specific Comments.

1. Claudin-low tumors are also characterized by increased immune and stromal cell infiltration. Is there any evidence of increased immune cells in mammary tumors from these oncogenic Kras models? Perhaps quantification of CD3 or other lymphocyte markers could address this.

2. The authors show the tumors in these models are basal-like and claudin-low and sometimes refer to them as triple-negative as in “luminal epithelial cells gave rise to claudin-low tumors in the oncogenic KRAS-driven, triple-negative breast cancer model.” They show two of the models are not HER2 amplified (Figure S2). However, the authors have not shown IHC staining for ER and PR in these models. Please provide IHC for ER and PR. This is especially important as claudin-low tumors have been demonstrated in other subtypes (PMID: 32286297).

3. How do the levels of p-ERK1/2 levels in the KRAS models compare to normal mammary glands in Figure S2?

4. Do any of the claudins (Cldn3, Cldn4 or Cldn7) increase in the derived cell lines when doxycycline is added?

5. Please include a scalebar for Figure 6E.

Reviewer #2 (Remarks to the Author):

This manuscript describes the analysis of three independent Kras-driven mouse models of mammary tumorigenesis. The authors hypothesize that aberrant Kras-signaling pathways in luminal mammary cells result can result in progressive differentiation and transformation leading to the claudin-low subtype of triple negative breast cancers. To test this possibility, they have generated three different Kras expression mouse models, using different orthologous genetic modifications, both transgenic-based and compound heterozygous-based transgenic/knockin models. Tumors generated from these models frequently express claudin-low transcriptional patterns, and the tumors are frequently metastatic after transplantation. The authors perform immunofluorescence to corroborate the transcriptional analysis and find that the tumors can express complex patterns of epithelial and mesenchymal markers that can alert over time. The authors interpret their data to suggest that like BRCA1 patients, claudin-low tumors may also arise from luminal tumor cells, rather than progenitor or stem cells.

This manuscript represents a significant amount of time, effort and resources to address this complex question in an in vivo animal model setting. While many of the interpretations may be supported by the data, there are a number of concerns that should be addressed to improve the manuscript:

- 1) The authors claim that the Kras-expressing tumors are “highly metastatic”. However, the only evidence of metastasis that is presented is from transplantation assays, not the primary autochthonous tumor. What percentage of the animals from which the original tumors were harvested had metastases? Are metastases only present after transplantation?
- 2) In the MMTV-tTA TetO-KrasG12D microarray analysis, the authors indicate that “four poorly differentiated carcinomas” were used. Were these tumors representative of the tumors in this population, or were they specifically selected because of their differentiation status?
- 3) Do the WAP-Cre and MMTV-tTA transgenes express in the same luminal cells? An alternative explanation for the different biology of tumors derived from these two transgenes is that they arise from different subpopulations of luminal cells. In addition, since the authors have demonstrated that the tumors are dependent on Kras expression, based on doxycycline treatment, it would suggest that the MMTV-tTA may be constitutively expressed like the Eef1a1 knockin.
- 4) In the Materials and Methods section, the authors indicate that the various knockin animals were backcrossed to wildtype FVB animals “up to seven times”. How many times was each of these models actually backcrossed? Since the authors are using compound heterozygotes for some of their analyses, significant heterogeneity could be introduced by inadequate backcrossing to clean up the genetic background.
- 5) For the Flp-inducible Kras experiments, the authors indicate that none of the “single transgenic littermates” developed tumors within a year. However, the experimental group is a triple, compound heterozygote, FSF-KrasG12D/p53R172H/MMTV-Flp. A more appropriate control would be the compound double heterozygous littermates, FSF-KrasG12D/p53R172H from the final cross. Do the authors have tumor incidence data on these animals?
- 6) There are significant concerns about the use of tissue culture conditions to assess the importance of constitutive Kras expression for the maintenance of mesenchymal phenotypes. As the authors indicate, transition from the in vivo to the in vitro system removes the requirement for Kras expression for tumor survival. Therefore, experimental results performed in tissue culture may be confounded by whatever adaptation tumor cells have undergone in the transition from the in vivo, tissue context, to the hyperoxic, hyper-rigid and nutrient-replete conditions in tissue culture.
- 7) Additional clarity should probably be provided for the assertion that Kras and Slug are acting “synergistically” as mediators of cellular plasticity. Based on the data shown in figure 7, Kras suppression results in an upregulation of E-cadherin in two out of the three cell lines. Knockdown of Slug however has inconsistent effects on E-cadherin expression. While there may in fact be an important interaction between Kras and Slug/Snail, the data, as presented, does not appear to fully support the author’s interpretation. Additional data, if available, would be helpful to support this hypothesis.

Reviewer #3 (Remarks to the Author):

The manuscript by Rädler and colleagues reports that persistent oncogenic RAS signaling causes highly metastatic triple-negative mammary tumors in 3 different mouse models. Noticeably, activation of KRAS in luminal cells in a continuous and differentiation stage-independent manner induces preneoplastic lesions that evolve into basal-like and claudin-low mammary cancers.

Overall, the manuscript is scientifically sound and of significant interest to the community. However, in its present form, it raises a number of points of discussion and concerns.

Major discussion points:

1) The entire study is focused on the consequences of RAS/MAPK signaling pathway on claudin-low tumor development. In the Introduction, this question is essentially based on the presence of hot-spot mutations in 3 out of 7 (42%) cell lines. Although recent publications dealing with this hypothesis have been addressed in the Discussion (Fonti et al., 2019; Fougner et al., 2020; Pommier et al., 2020), they need to be referred to in the introduction to strengthen the relevance of the question.

2) The authors have generated 3 transgenic mouse models.

In the first model, MMTV-TetOKRas/tetOGFP, KRAS is expressed in CK8-positive cells (the majority of cells) and in very few CK14-positive cells, the tumors are poorly differentiated with selected areas of local invasion. The number of mice is relatively low (n=8 for the control and n=9 for the activated KRAS). Additional information is lacking such as tumoral growth curve and tumoral penetrance.

3) The histological examination of the tumors reveals a certain heterogeneity with selected areas of local invasion and EMT.

The EMT is not characterized in detail. The authors describe “spindle-shaped cells” that could not be differentiated from fibroblasts without the use of a proper characterization.

4) The authors perform an orthotopic transplantation of small tumor fragments of dissociated cancer cells into WT mice and the secondary tumors resembled those of the primary tumors (“selective enrichment of EMT-like cells in transplants that originated from a cancer that had more extensive areas of spindle-shaped tumor cells (Fig. 1C, right). However, there is no quantification of

the proportion of “spindle-shaped cells”. It would also be interesting to observe secondary tumors originated from different areas of the same primary tumors, i.e. is the heterogeneity of the tumor reproduced or is there a certain degree of plasticity in the secondary tumor?

5) The transplanted tumors are associated with the formation of metastases in the lungs. Were other sites analyzed (liver, bone)? Are there any metastases in the tumors from the MMTV-TetOKRas/tetOGFP model?

6) Doxycycline treatment eliminates KRAS expression and signaling, and there is regression of the tumor with the presence of residual GFP reporter cells at the initial tumor site and metastasis. The illustration of the initial tumor is lacking. The authors hypothesize that the residual cells are quiescent cells. A Ki67 staining would be necessary to validate this hypothesis.

What is the fate of these quiescent cells? Do they show a tumor formation capacity, would they become RAS-independent?

7) Gene expression analysis was performed on 4 poorly differentiated carcinomas that originated in MMTV_tTA TetO-Kras G12D mice. The authors use a transcriptomic microarray for the first two transgenic models. We are surprised by the use of this technology as more advanced technologies (i.e. RNAseq) are currently used for transcriptomic analyses.

Only one of the 4 tumors was presenting more extensive EMT characteristics. Is that ratio a general observation of all the tumors?

Only 4 out of 9 tumors have been characterized. Additional characterization of the EMT in the 5 remaining tumors would be appreciated.

The authors correlate the expression data to those published in Herschkowitz et al, 2007: “The results showed that three KRASG12D-driven carcinomas clustered to the Class 8 tumor-type and were similar to cancers that originated in MMTV-HRAS, WAP-Myc and WAP-Int3 transgenic females”. Is “Class 8” the same as Group VIII described in Herschkowitz et al, 2007?

8) The second murine model developed in this study is WAP-Cre EF1-LSL-tTA TetO-KrasG12D.

This model is highly relevant for the study and it brings originality and robustness to the manuscript.

“Since most cancer cells in the MMTV-tTA TetO-KrasG12D model are dependent on the continuous expression of the oncogene, it is likely that the ability of luminal-type cancer cells to transdifferentiate into a complete basal-like and mesenchymal-like state is restricted.”

Is this statement based on previous studies? There is a lack of references supporting this notion.

9) The authors describe “the development of palpable mammary tumors”. A growth curve of the tumor formation rather than tumor-free survival is needed to illustrate the latency. The penetrance of the model needs to be indicated.

10) The microarray-based gene expression analysis revealed that all mammary tumors clustered to the claudin-low intrinsic subtype. How many tumors were analyzed, including the adenocarcinoma?

11) Using the Cre/lox methodology as a cell-lineage tracing tool, the authors characterize the EMT program at the early stages of tumor initiation with a particular focus on parity-induced mammary epithelial cells (PI-MECs) where the WAP promoter is activated. The majority of PI-MECs are luminal-type cells and distinct from CK14-positive basal epithelial cells. Oncogenic KRAS triggered an expansion of CK8-positive PI-MECs within focal regions of the gland, but more interestingly, isolated cells with typical luminal-type appearance started to co-express CK14 and become E- and N-cadherin-positive.

A better characterization of EMT is needed and expression of other markers such as occludin, ZO-1, EMT-TFs needs to be addressed.

12) A third model developed in the study is MMTV-Flp FSF-KrasG12D p53R172H where mutant KRAS is expressed from its endogenous locus in a Flp recombinase-inducible manner and associated with a single mutant p53R172H allele.

We are unsure about the choice of MMTV promoter for this model rather than the WAP system given that the authors explain the limits of MMTV promoter in the paragraph “A luminal epithelial cell-specific activation of oncogenic RAS under the control of a ubiquitously

active promoter leads to the development of highly metastatic, claudin-low mammary cancer”.

13) Mammary tumors were not observed in any of the single transgenic littermate controls within one year. The number of mice is quite low, n=6 for Ras, n=12 for control, and n=9 for mutant p53. We are surprised by the fact that the single transgenic RAS mice did not develop tumor whereas in the previous models, expression of RAS alone is sufficient for tumor formation. Is it due to a different level of KrasG12D expression? A higher number of mice in the Ras group would increase the statistical power of the study.

14) The oncogenic activation of endogenous KRAS associated with mutated p53 caused EMT-like, spindle cell tumors and poorly differentiated adenocarcinomas. The authors analyzed 3 tumors of each subtype. KRAS protein levels in tumors that originated in the EF1-tTA-based transgenic model were similar to those tumors that express mutant KRAS from the endogenous locus. A quantification of the signal detected by WB is lacking, and a higher number of tumors need to be analyzed.

15) RNAseq analysis revealed that the poorly differentiated adenocarcinomas exhibited molecular features of the basal-like subtype, and the spindle-shaped, mesenchymal-like tumors clustered closely to the claudin-low reference sets. The authors characterize the loss of E-cadherin, EpCAM and decrease in CK14 in CL tumors, while basal-like tumors retain the expression of CK14, EpCAM and E-cadherin. A better characterization by FACS analysis of the percentage of EpCAM-positive cells is needed.

16) The authors observe that the tumors retained expression of Cdkn2a and the p19Arf protein (Suppl. Fig. S7). This might suggest that the loss-of-function of Cdkn2a precedes the acquisition of p53 mutations. Analysis of the loss of Cdkn2a, and p53 mutations in TCGA or METABRIC datasets would be more relevant than in MDA-MB 231, Hs578T and SUM159PT, for which there are no studies cited in the manuscript.

17) The authors perform a RNA-Seq experiment on three cell lines to determine resulting changes in their transcriptomes when the expression of oncogenic KRAS was terminated.

The purpose of this experiment is unclear. Is it to validate the maintenance of the CL status or to identify RAS-activated signaling pathways?

18) The authors demonstrate the changes in the transcriptional activation of key markers, in particular CD24, CD49f (Itga6), CD61 (Itgb3) by flow cytometry, and reversion of the phenotype from CD49fhigh/CD24-/low/CD61- to CD49f+/CD24+/CD61low after RAS extinction.

“This redifferentiation process towards a more luminal epithelial fate of these tumor cells is accompanied by a substantial increase in the number of cells that exhibit high levels of CD24 and a marked reduction in CD49f (Fig. 7A, 7C). The partial reversal of the transdifferentiation program supports our findings that luminal epithelial cells gave rise to claudin-low tumors in the oncogenic KRAS-driven, triple-negative breast cancer model.”

Control experiments in luminal or CL cell lines are necessary to validate this conclusion.

The time points 48h, 72h or 96h are not indicated and discussed in the text.

We wonder about the pertinence of the conclusion. It is not because “claudin-low” tumor cells exhibit a luminal phenotype after RAS extinction that they came from luminal cells. This conclusion does not seem to be founded on results presented.

19) The authors investigate if the suppression of RAS leads towards a MET process. The conclusion that there is an incomplete MET is based solely on E-cadherin expression analysis (because N-cadherin expression remains unchanged). It is necessary to analyze other markers such as vimentin, EpCAM, ZO-1, Occludin) and other EMT transcription factors (ZEB1/2, TWIST1/2).

20) The authors observe that the suppression of oncogenic KRAS leads to a substantial increase in the level of the SLUG protein. Slug inactivation by shRNA leads to the re-expression of E-cadherin in 2 out of 3 cell lines. However, there is a visible re-expression of E-cadherin in only 1 cell line in the western blot analysis (quantification needed) and this increase in E-cadherin is associated with a decrease in the expression SNAIL (a well-known negative regulator of E-cadherin). Therefore, the conclusion that knockdown of SNAIL is sufficient to upregulate E-cadherin expression in the absence of oncogenic KRAS (sup. Figure S8B) is misleading.

Overall, these experiments do not deal with cellular plasticity. What are the analyses that allow the authors to conclude that “SLUG and SNAIL act synergistically with oncogenic RAS signaling as mediators of cellular plasticity”?

21) “Nonetheless, the continuous repression of E-cadherin in one of the three tumor cell lines (line 2) in the absence of oncogenic KRAS, SLUG, and SNAIL implies that, despite significant increases in ITGB3 and GATA3, the maintenance of mesenchymal characteristics of claudin low mammary cancer cells is multifactorial and can be controlled by additional molecular mechanisms.”

The analysis of other transcriptional factors inducers of EMT (i.e. ZEB1/2, TWIST1/2) is required.

22) There is a lack of perspectives in the discussion concerning previously published murine models of mammary tumor initiation in the context of RAS, such as publications by Jane Visvader lab, Cédric Blanpain lab or Harold Varmus lab. What is the novelty and advantages of the three newly proposed models compared to previously published ones?

Minor discussion points:

1) Tumors showed a transcriptional repression of the Cdkn2a locus, the resulting lack of the p19Arf and p16Ink4a protein expression, and, in some cases, additional sporadic mutations in Trp53. A better characterization of p53 mutation would be appreciated. Sequencing analysis should be done in addition to the western blot analysis.

2) The authors identify MAPK signaling pathway genes, ECM-receptor interaction genes, stem cell signaling genes and mammary lineage markers (CD24a, CD61, CD49f et GATA3). References would be appreciated.

3) In the paragraph, "The cellular plasticity is controlled by persistent oncogenic RAS signaling", we do not fully understand the rationale in analyzing Caspase-9 cleavage. Authors should clarify this point.

Response to Referees

The authors would like to thank the reviewers for taking time from their busy schedules to thoroughly read our manuscript. We are pleased that the reviewers felt that *'the results are scientifically sound and of significant interest to the community'* (reviewer #3). Our work provides *several novel mouse models and insight into the cellular origin of claudin-low breast cancer* (reviewer #1), and reviewer #2 is certainly correct to point out that this manuscript represents a *significant amount of time, effort, and resources to address this complex question in an in vivo animal model setting*. In response to the suggestions, we invested a significant amount of time and resources to perform additional experiments during the COVID19 pandemic. Consequently, the number of composite figures of this manuscript has increased to 21 (7 main and 14 supplemental figures). In this correspondence, we will summarize, in a point-by-point manner, how we acknowledged and addressed the issues that were stated and answer additional questions.

Reviewer #1

1. *"...Is there any evidence of increased immune cells in mammary tumors from these oncogenic Kras models? Perhaps quantification of CD3 or other lymphocyte markers could address this?"*

As requested, we analyzed the presence of CD3-positive immune cells in both basal-like and claudin-low mammary tumors that originated in the same genetic model expressing mutant KRAS from its endogenous locus. The data shown in the new Suppl. Fig S11 is on par with previous reports that there is an elevated presence of immune cells in claudin-low mammary cancers.

2. *"The authors show the tumors in these models are basal-like and claudin-low and sometimes refer to them as triple-negative... Please provide IHC for ER and PR...."*

In response to the reviewer's comment, we performed immunoblot analyses on claudin-low mammary tumors expressing exogenous mutant KRAS (new Suppl. Fig S2A) and the mutant KRAS protein under the control of the endogenous KRAS locus (new Suppl. Fig S8). As controls, we used lysates from tumors that were classified as basal-like and samples from MMTV-neu-induced tumors, and we also included mouse uterine tissues as positive controls for ER α and PR (i.e., PR-B, the positive regulator of progesterone signaling). The collective results confirm that basal-like and claudin-low mammary cancers in our two genetic models do not express ER α and PR, nor do these tumors exhibit an amplification or overexpression of ERBB2.

3. *"How do the levels of p-ERK1/2 levels in the KRAS models compare to normal mammary glands in Figure S2?"*

It seems less informative to run immunoblots for a comparison of pERK1/2 between the normal glands of wildtype controls and transgenics or between normal glands and tumors due to differences in the cellular composition. The elevated levels of pERK1/2 in transforming epithelial cells compared to normal mammary epithelial cells are more accurately illustrated by immunofluorescent staining as shown in the new Supplemental Fig S4. The higher phosphorylation levels of MAP kinases are clearly linked to the Cre-mediated activation of the CAG-GFP reporter, the EF1-tTA, and the consequential expression of the TetO-KRAS^{G12D} transgene. The GFP-positive premalignant cells and cancer cells within a large tumor show the same intensity in pERK1/2 signals.

4. *“Do any of the claudins (Cldn3, Cldn4, or Cldn7) increase in the derived cell lines when doxycycline is added?”*

The expression of these claudins was not significantly higher in the three cell lines when treated with Dox despite higher levels of GATA3, CD24, and CD61, which supports our conclusion that the mesenchymal-epithelial transition (MET) process was incomplete in the absence of oncogenic RAS signaling as shown and discussed in the final part of the Results section.

5. *“Please include a scalebar for Figure 6E.”*

Scale bars were included. Thanks for noticing.

Reviewer #2

1. *“...What percentage of the animals from which the original tumors were harvested had metastases? Are metastases only present after transplantation?”*

Pulmonary metastases were also found in primary tumor-bearing mice of all three models. We have included the information in the revised manuscript. Specifically, four out of nine (45%) mice of the MMTV-tTA TetO-Kras model on which we performed a complete necropsy had overt lung metastasis. Similarly, 13 of 33 (40%) WAP-Cre EF1-tTA TetO-Kras had macro-metastases at the time of necropsy, and among those lung lesions, 12 were GFP-positive in mice that carried the CAG-GFP reporter. The propensity of primary tumors to metastasize is likely higher since we were not allowed to keep any females alive with tumors that had reached a size of 1.5 cm in diameter (see M&M section) as mandated by the University’s Animal Use and Care Committee. Lung metastases were also observed in female mice that developed mammary tumors in response to the Flp-mediated activation of the endogenous mutant *Ras* allele as we have shown in the bottom panel of Fig. 5B.

2. *“...In the MMTV-tTA TetO-KrasG12D microarray analysis...were these tumors representative of the tumors in this population, or were they specifically selected because of their differentiation status?”*

In this model, all mammary tumors were quite similar and were classified by our pathology expert, Bob Cardiff, as poorly differentiated large-cell carcinomas. As described in the manuscript, there was some variation within selected areas of these tumors that show local invasion and cells with mesenchymal properties. Nonetheless, except for one tumor, all others show close similarities to the classic MMTV-HRAS model in their expression profiles. We believe the reason for this is that the vast majority of MMTV-driven tumors retained luminal-type characteristics (i.e., CK8 expression) despite a gain of EMT-like appearances and expression of basal-type keratins, as shown in Figure 1. Unfortunately, we did not retain viable tumor cells from the single tumor that clustered to the claudin-low type in this model to assess whether cancer cells had escaped from their dependency on the MMTV-driven expression of the tTA or even their dependency on mutant *Kras*. It is our collective observation on several transgenic models that *the MMTV-LTR is not expressed in mesenchymal cells* of the mammary gland. Hence, it is unlikely that the MMTV-LTR is constitutively active in cancer cells that have completed a mesenchymal transition while retaining their dependency on the MMTV-driven expression of the oncogene (here KRAS).

3. *“...Do the WAP-Cre and MMTV-tTA transgenes express in the same luminal cells? An alternative explanation for the different biology of tumors derived from these two transgenes is that they arise from different subpopulations of luminal cells.”*

There is a significant overlap in the expression profiles of the WAP-Cre and MMTV-tTA. In fact, we have previously demonstrated that oncogenes under the control of the same MMTV-LTR (i.e., MMTV-neu) led primarily to mammary tumors within “parity-induced mammary epithelial cells (PI-MECs)” that were labeled with WAP-Cre and the ROSA-LSL-LacZ reporter (Henry et al. 2004 Oncogene 23 (41): 6980–6985). More importantly, and as demonstrated in this manuscript, we used the same MMTV-LTR to generate the MMTV-tTA and MMTV-Flp mice, and the expression of mutant KRAS in the same cell types resulted in completely different tumor types due to differences in the mode of expression of the oncogene (i.e., MMTV-tTA-dependent versus MMTV-LTR independent in the MMTV-Flp-based model). Just the comparison between these two models shows that the differences in the molecular tumor subtypes are not a result of an expression of oncogenic RAS in dissimilar cell types. We also have shown that these differences are not caused by hypothetical variations in KRAS expression. Therefore, the major differences in the gene expression profiles are most likely caused by the differences in the mode of oncogene expression, i.e., KRAS functionally tethered to the MMTV-LTR expression in the first model versus the expression of KRAS under ubiquitous promoters that are differentiation-independent (EF1-tTA or endogenous KRAS in the MMTV-Flp-based model).

4. *“...Since the authors are using compound heterozygotes for some of their analyses, significant heterogeneity could be introduced by inadequate backcrossing to clean up the genetic background.”*

We keep electronic records of the genealogy of every single animal in our colony, its primary source, genotypes, genetic background, and their offspring since 1999. The MMTV-tTA and MMTV-FLP lines were generated and always maintained in the FVB background, which is of paramount importance for the expression of the MMTV-LTR and the initiation of MMTV-LTR-driven/induced tumors. As stated in the M&M section, the latest additions to our colony (i.e., FSF-KRAS and the mutant p53 alleles) were backcrossed independently more than seven times into wildtype FVB. The actual FVB content is likely higher when these mice were mated with the MMTV-Flp mice. In a future publication, we will show the ability to transplant mutant KRAS-driven tumor cells into wildtype, immunocompetent FVB recipients that develop cancer in as little as 11 days. This would not be possible if we had significant heterogeneity in the genetic background.

5. *“...For the Flp-inducible Kras experiments, the authors indicate that none of the “single transgenic littermates” developed tumors ...a more appropriate control would be the compound double heterozygous littermates, FSF-Kras^{G12D}/p53^{R172H} from the final cross. Do the authors have tumor incidence data on these animals?”*

The FSF-Kras^{G12D} allele is a functional knockout of KRAS and has no phenotypic contribution to whatever a heterozygous p53^{R172H} allele might have caused. There is no spontaneous recombination between FRT sites in the absence of Flp recombinase in the mammary gland or elsewhere. We have maintained seven FSF-Kras^{G12D}/p53^{R172H} breeder females over the years that lacked the MMTV-Flp. None of these animals developed mammary tumors within approximately one year (377 days). We have included this information in the text.

6. *“...experimental results performed in tissue culture may be confounded by whatever adaptation tumor cells have undergone in the transition from the in vivo, tissue context, to the hyperoxic, hyper-rigid and nutrient-replete conditions in tissue culture”*.

It is important to recognize the differences in the dependency of cancer cells on oncogenic RAS when grown in culture and *in vivo* that we show in this work. This phenomenon is not unique to our cell culture models and applies to pancreatic cancer cells where the vast majority of tumors

are dependent on mutant RAS-signaling *in vivo* but not *in vitro* (PMC5369772, PMC6888344). However, the adaptation process for cells to grow in culture and to survive *in vitro* without mutant KRAS did not change their dependency on this driver oncogene when the cells are subsequently transplanted back into mice. The secondary tumors will regress when the mice are being treated with doxycycline to suppress the expression of the driver oncogene as demonstrated in several of our previous publications (PMC3992817, PMC5369772). Therefore, an adaptation process that promotes the growth of cancer cells in culture does not fundamentally change important KRAS signaling circuits. There is an experimental upside to the phenomenon that cancer cells can survive in culture without mutant KRAS. Without sustained survival, it would have been nearly impossible to I) study the role of the driver oncogene and additional molecular pathways that control cellular plasticity, and II) provide additional evidence that claudin-low mammary tumors have likely originated from luminal epithelial cells.

7. "...Additional clarity should probably be provided for the assertion that *Kras* and *Slug* are acting "synergistically" as mediators of cellular plasticity. ...Additional data, if available, would be helpful to support this hypothesis".

We agree that the word 'synergistically' does not accurately reflect the relationship between KRAS and SLUG in the maintenance of cellular plasticity in claudin-low mammary cancer, and we, therefore, removed this statement. SLUG was consistently upregulated in all cell lines when the expression of mutant KRAS was turned off. Suppression of the increase in SLUG when the expression of mutant KRAS was ablated resulted in a further increase in E-cadherin in 1 out of 3 cell lines. In another cell line where suppression of oncogenic KRAS alone led to much higher upregulation of E-cadherin, the inhibition of SLUG had no additional effect. During the revision of this manuscript, we also analyzed the expression of other classic EMT factors (ZEB1 and TWIST) in addition to SLUG and SNAIL (see new Suppl. Fig S13). The results and conclusions remained the same. SLUG is the only one of these factors that is consistently upregulated on the protein level when KRAS is turned off. Whether SLUG acts along with any of the other EMT factors is the subject of future investigations. Nonetheless, what the collective data in this manuscript shows is that the degree of cellular plasticity is dependent, in a significant part, on the continuous expression of the main oncogenic driver and that other molecular pathways may contribute to this developmental process in an oncogene-independent manner.

Reviewer #3

Reviewer #3 stated that our work is scientifically sound and of significant interest to the community. We carefully considered every suggestion and conducted additional experiments, whose results are shown in the new Supplemental Figures S5, S9, and S13. To keep the response as succinct as possible, we reorganized some of the comments that were thematically connected.

Comment 1. As requested, we cited the papers by Fougner et al. (2020) and Pommier et al. (2020) in the *Introduction* section that mentioned the role of RAS signaling in claudin-low mammary cancer.

Comments 2 through 7 were related to our introductory mammary cancer model (model 1) where KRAS is expressed in a classical fashion under the MMTV-LTR, albeit in a tTA-controlled manner. The collective results shown in Figs. 1 and 2, confirm that the traditional transgenic approach of expressing KRAS tethered to the MMTV promoter (here MMTV-tTA) resulted in mammary tumors

that are histologically and molecularly similar to earlier published models expressing different RAS isoforms under mammary gland-specific promoters. The sole purpose of model 1 was to highlight the innate inability of the classic transgenic approach to model advanced tumor cell plasticity while retaining a dependency on the transforming oncogene. The MMTV-LTR is not active in mesenchymal cells (see Supplemental Figures S1 and S6). Given the word and character limitations of this commentary-style article, we abstained from conducting further studies on model 1 and did not deliberate details about experimental modeling in the *Discussion* section. We kept the primary focus of this work on the models 2 and 3, which as correctly stated in *comment 8*, highlight the novel concept and findings reported in this manuscript.

The tumor penetrance for model 1 is 100% and illustrated in Fig 1A (*comment 2*). The expression of luminal and basal keratins (Fig 1B, lower panels, gray arrows) is a clear indication that the spindle-shaped cells are tumor cells and not fibroblasts (*comment 3*). Since the MMTV-LTR is not expressed in mesenchymal cells of the normal or neoplastic mammary gland, it is evident that, under normal circumstances when the survival of tumor cells is dependent on the MMTV-driven expression of the oncogene, there is no selective enrichment for cells that undergo extensive EMT in this model (*comment 4*). Such a process would only happen if the growth and survival of tumor cells would become independent from the functionality of the tTA system. This may have been the case in the single tumor shown in the manuscript that showed an enrichment of spindle-shaped cells after transplantation. Unfortunately, we did not have different areas from the same tumor for transplantation since primary tissues were also processed for histology and protein expression analyses. The histopathologic features of a tumor are only available *after* the specimen was fixed, sectioned, and stained. There is no practical way to assess the presence of different amounts of metaplastic cells within viable tumor fragments that are about 1 mm in size prior to transplantation.

Other than pulmonary metastases, we did not see any other sites or organs of metastatic dissemination in model 1 (*comment 5*). For a general description of the fate of residual cancer cells that survive the ablation of KRAS (*comment 6*) and other oncogenes and their ability to give rise to secondary tumors please see our earlier papers (PMC5369772, PMC3992817). The proposed experiment to further investigate the dormant cancer cells in model 1 is beyond the scope of this manuscript. Regarding the proliferative capacity of the dormant cancer cells, it is likely that the residual cancer cells (Fig 1D) do not multiply since they retain the nuclear expression of the H2B-GFP despite Dox-mediated suppression of the TetO-H2B-GFP transgene. The concept of using the TetO-H2B-GFP transgene to genetically label quiescent stem cells in the skin is described in detail in reports from the laboratory of Elaine Fuchs, and we have utilized this methodology previously to label dormant cancer cells (PMC3992817, PMC5369772).

Authors of this manuscript have been pioneering array-based and RNA-seq-based methodologies to cluster human breast cancers as well as mouse mammary tumor models based on their molecular profiles. It might not be correct to refer to the array technology as inferior (*comment 7*). It took many years to generate all three new genetic cancer models and to gather the experimental data presented in this manuscript. The computational methodologies to use RNA-seq for the transcriptomic-based molecular classification and more importantly, the RNA-seq data sets from validated mouse mammary cancer models for comparison became available only recently. We were able to apply the new technologies to the latest, MMTV-Flp-based claudin-low mammary cancer model.

Comment 9: Similar to model 1, the tumor penetrance in model 2 is 100% and shown in Fig. 3A. This graph also illustrates the cancer latency in response to the EF1-tTA-mediated expression of TetO-KRAS as described in the manuscript. Tumor penetrance and latency provide no information about variations in the growth rates of individual tumors once they have been detected. The mammary cancers in our models at the time of necropsy are standardized by the size of the tumor (1.5 cm as the allowed biological endpoint by the IACUC as described in the manuscript). All tumors in model 2 at the biological endpoint were claudin-low. Therefore, any variation in the growth rates of individual tumors to reach the allowed maximal size are not determinants for the histopathological or molecular characteristics of these mammary cancers.

Comment 10: The mammary tumor model 2 develops only carcinomas, and the numbers are described in the Results section.

Comment 11: As requested, we performed an analysis of additional EMT markers, including Occludin and EMT-TFs. While nuclear expression of SLUG and SNAIL could not be detected in premalignant lesions (and therefore is not shown), we demonstrate in the new Supplemental Fig S5, that Occludin is lost in transforming cells that have gained N-cadherin as opposed to adjacent epithelial cells in the same field of view that have not initiated an EMT program. Considering that N-cadherin-positive epithelial cells are strongly dual positive for luminal and/or basal keratins, it should be evident that the EMT process in these cells is far from being complete.

Comment 12. A transactivator-mediated expression of a transgene is fundamentally different from a recombinase-mediated activation of an endogenous locus. The MMTV-tTA needs to be expressed continuously to transactivate the KRAS responder transgene. Hence, the KRAS transgene perpetually remains under the control of the MMTV-driven, and therefore differentiation-dependent expression of the tTA. The same MMTV-LTR that was used in model 1 drives the expression of the Flp recombinase in model 3. The MMTV-mediated expression of Flp will lead to the activation of the mutant *Kras* allele. Once Flp has completed its sole function to recombine the Stop sequence flanked by FRT sites, the expression of mutant KRAS is under the control of its own ubiquitously active promoter whether the MMTV-Flp remains active or is turned off during metaplastic transition. In contrast to the MMTV-tTA, the MMTV-Flp-mediated expression of mutant KRAS is differentiation-independent in model 3. Therefore, using the same MMTV promoter seemed to be the perfect choice to demonstrate that the mode of activation of the same mutant oncoprotein is a primary determinant for the cellular plasticity and consequential change of the molecular cancer subtype.

Comment 13: As described above, the activation of the mutant *Kras* allele in model 3 relies on the temporary functionality of the MMTV-Flp. Without the Flp transgene, the *FSF-Kras^{G12D}* gene is a complete null mutant of KRAS. Mice carrying just the *FSF-Kras^{G12D}* therefore never develop cancer. We have also adequately assessed the expression of mutant KRAS and downstream activation of ERK among all three models. In total, we compared the protein expression of 15 individual RAS-driven tumors from the three models as well as three tumors from ERBB2 overexpressing mice (Figs. S2 and 5C). Some variations in KRAS expression and ERK activation between the three models are no different than those observed between individual tumors of each of the models. Hence, our data show that the genesis of the luminal, basal, or claudin-low molecular subtypes are not linked to any differences in the levels of the mutant oncoprotein or activation of ERK. There is no trend in the data to suggest further examination beyond the 15 individual KRAS expressing cancers that we analyzed by immunoblot.

Comment 15: As requested, we have reassessed the expression of EpCAM on the level of single cells. We now show a comparison of EpCAM expression on immunofluorescent-stained section of tumors of both subtypes (basal-like and claudin-low) in the Supplemental Figure S9.

Comment 16 and minor discussion point 1: The suggestion to use METABRIC data for building a hypothesis and to further investigate the relationship between mutations in *p53* and *CDKN2A* are well taken, but these lines of investigations are beyond the scope of this manuscript. Thank you for suggesting interesting future lines of investigation.

Comment 17: Conducting RNA-seq experiments on Dox-treated mammary cancer cells served multiple purposes, which included getting a global picture of the gene expression profiles that are driven by RAS and selected pathways and molecular features associated with cellular plasticity and stemness as described in this manuscript. This RNA-seq experiment will be the basis for additional reports in the future.

Comment 18: For the flow cytometric analyses, we used all necessary controls for each individual antibody used. These antibodies have been repeatedly validated on a very wide variety of mammary cancer models, including luminal-type tumors. We provide the citations of relevant articles (see also *minor comment 2*). Primary flow data on luminal-type MMTV-neu tumors as controls are also provided in one of our recent publications (Wehde et al., 2018 in *Cell Reports*). The duration of the Dox treatment for the results shown in this manuscript is described in the figure legends. The finding that claudin-low cells upregulate luminal epithelial marker following the downregulation oncogenic RAS should be viewed as additional evidence that claudin-low cancer cells came from luminal progenitors of their more differentiated descendants. This statement is based on several lines of evidence presented in this paper, including the genetic labeling using the Cre/lox system and images showing the transitional states of preneoplastic mammary epithelial cells that have characteristics of luminal and basal cells and that show signs of EMT progression such as expression of N-Cadherin. In summary, evidence from three independent lines of investigation was presented regarding the likely origin of claudin-low mammary cancer in our novel mammary tumor models.

Comment 19: In addition to E- and N-cadherin, we conducted immunoblots for EpCAM and Occludin that confirmed that the latter two are not expressed. Their CPM values were below 1 in the RNA-seq data regardless of Dox-treatment as reported in the revised manuscript. We also analyzed potential changes in the expression of other EMT transcription factors and conducted immunoblots for ZEB1 and TWIST as requested. The collective results shown in Fig. 7 and the new Supplemental Fig. S13 confirm our statement that the MET process is incomplete following the downregulation of oncogenic RAS.

Comment 20: Changes were made in the manuscript regarding the former statement of a synergistic role of KRAS and SNAIL in cellular plasticity. Unless changes in the expression of a protein occurred in a similar manner in all three cell lines (e.g., SLUG), the differential effects of the downregulation of mutant KRAS and shRNA-mediated knockdown of SNAIL or SLUG were described for the individual cell lines.

Comment 21: The requested analysis of ZEB and TWIST was discussed in *comment 19*. The immunoblot data is shown in the new Supplemental Fig. S13B. The results confirm our hypothesis that the maintenance of mesenchymal characteristics of claudin-low mammary cancer cells is multifactorial and can be controlled by additional molecular mechanisms.

Comment 22: To our knowledge, there has never been any study thus far that directly compared the biological consequences of expressing the same oncogene using entirely different approaches, and no studies have directly shown that the cellular and molecular characteristics of cancer cells are dependent on the mode of oncogene expression. The basic idea for this novel concept was published in the discussion section of our recent technical article in the journal *Scientific Reports* (Sakamoto et al., 2020), which is cited in this new manuscript. The first paragraph in the second section of the *Results* in this report describes the rationale and methodology of the new approach. Given that this journal has strict limitations in the number of words and characters, we were unable to expand the *Discussion* section and describe further technical details and conceptual advances in mammary cancer modeling. This information could be provided as part of a methodology review in the future.

Minor comments:

1) see the response to *comment 16*.

2) As requested, citations were included that refer to the primary literature describing the identification and use of epithelial lineage markers.

3) We showed the cleavage of Caspase-3 (or lack thereof in Dox-treated cells) to provide evidence that cells are not apoptotic when the expression of mutant RAS is turned off. In fact, the downregulation of mutant RAS caused less oncogene-induced cell stress and death in culture as mentioned in the manuscript.

REVIEWERS' COMMENTS:

Reviewer #1 (Remarks to the Author):

The authors have provided additional experimentation that have satisfied all my concerns and have sufficiently addressed all the reviewers comments.

Reviewer #2 (Remarks to the Author):

The authors have adequately addressed the majority of my concerns. However, the issue regarding the compound heterozygous nature of the animals remains to be resolved. In the text of the manuscript, the authors state that the new mice were "were backcrossed up to seven times with FVB/N wildtype mice." This indefinite language can mean that some animals were backcrossed once, while others were backcrossed seven times. Obviously this would result in potentially different fractions of donor versus FVB genomes in the experimental animals. In addition, in the rebuttal however, the authors state that they were backcrossed "more than seven times." The ambiguity about the number of backcrosses and the discrepancy between the manuscript and rebuttal therefore still needs to be resolved.

Please note that the simple request to document the number of backcrosses performed for each model imported was not an implied criticism of the authors expertise or record keeping, but was made to improve the clarity of the manuscript and remove ambiguity from the experimental methods. The data regarding the transplantability of the tumors into naive FVB described in the rebuttal is not presented in this manuscript and therefore cannot be used as a validation of the genomic similarities between the models in question. The authors can resolve this issue by simply adding the number of backcrosses for each to the materials and methods.

Reviewer #3 (Remarks to the Author):

The authors have sufficiently addressed most of our concerns and significantly improved the manuscript in this revised version.

We do not wish to further comment the unnecessary and unpleasant remarks.

Response to Referees

The authors would like to thank all three reviewers for taking time from their busy schedules to reassess our work and responses to previous comments that have helped to improve the quality of the manuscript. We are pleased that the reviewers felt that we addressed most of the issues that were raised in a satisfactory manner and that the manuscript is, in principle, suitable for publication. As requested by reviewer 2, we have included the number of backcrosses of the mutnat mouse strains in the Material & Methods section of the manuscript.